# MotionCLR: Motion Generation and Training-free Editing via Understanding Attention Mechanisms

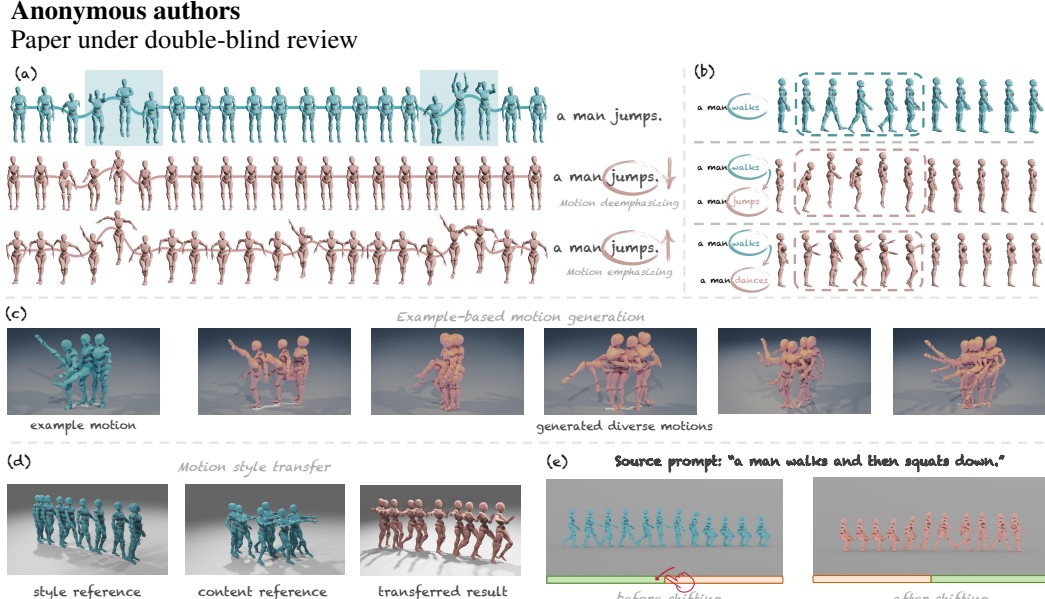

Figure 1: **MotionCLR supports versatile motion generation and editing.** The blue and red figures represent original and edited motions. (a) Motion deemphasizing and emphasizing via adjusting the weight of "jump". (b) In-place replacing the action of "walks" with "jumps" and "dances". (c) Generating diverse motion with the same example motion. (d) Transferring motion style referring to two motions (style and content reference). (e) Editing the sequentiality of a motion.

## ABSTRACT

This research delves into the problem of interactive editing of human motion generation. Previous motion diffusion models lack explicit modeling of the word-level text-motion correspondence and good explainability, hence restricting their fine-grained editing ability. To address this issue, we propose an attention-based motion diffusion model, namely MotionCLR, with CLeaR modeling of attention mechanisms. Technically, MotionCLR models the in-modality and cross-modality interactions with self-attention and cross-attention, respectively. More specifically, the self-attention mechanism aims to measure the sequential similarity between frames and impacts the order of motion features. By contrast, the cross-attention mechanism works to find the fine-grained word-sequence correspondence and activate the corresponding timesteps in the motion sequence. Based on these key properties, we develop a versatile set of simple yet effective motion editing methods via manipulating attention maps, such as motion (de-)emphasizing, in-place motion replacement, and example-based motion generation, *etc*. For further verification of the explainability of the attention mechanism, we additionally explore the potential of action-counting and grounded motion generation ability via attention maps. Our experimental results show that our method enjoys good generation and editing ability with good explainability. Codes will be public.

## 1 INTRODUCTION

Recently, text-driven human motion generation (Ahn et al., 2018; Petrovich et al., 2022; Tevet et al., 2022b; Lu et al., 2024; Guo et al., 2024a; Hong et al., 2022; Wang et al., 2022; 2024) has attracted significant attention in the animation community for its great potential to benefit versatile downstream applications, such as games and embodied intelligence. As the generated motion quality in one inference might be unsatisfactory, interactive motion editing is valued as a crucial task in the community. To provide more interactive editing capabilities, previous works have tried to introduce

some human-defined constraints into the editing framework, such as introducing a pre-defined trajectory for a controllable generation (Xie et al., 2024a; Dai et al., 2024; Shafir et al., 2024) *or* key-frames for motion in-betweening (Chen et al., 2023a; Tang et al., 2022; Harvey et al., 2020).

Despite such progress, the constraints introduced in these works are mainly in-modality (motion) constraints, which *require laborious efforts in the real animation creation pipeline*. Such interaction fashions strongly restrict the involving humans in the loop of creation. In this work, we aim to explore a natural editing fashion of introducing out-of-modality signals, such as editing texts. For example, when generating a motion with the prompt "a man jumps.", we can control the height or times of the "jump" action via adjusting the importance weight of the word "jump". Alternatively, we can also *in-place* replace the word "jump" into other actions specified by users. Moreover, in this work, we would like to equip the motion generation model with such abilities without re-training.

However, the key limitation of existing motion generation models is that the modeling of previous generative methods lacks **explicit word-level** text-motion correspondence. This fine-grained cross-modality modeling not only plays a crucial role in text-motion alignment, but also makes it easier for fine-grained editing. To show the problem, we revisit previous transformer-based motion generation models (Tevet et al., 2022b; Zhang et al., 2024b; 2023b; Chen et al., 2023b). The transformer-encoder-like methods (Tevet et al., 2022b; Zhou et al., 2024) treat the textual input as one special embedding before the motion sequence. However, text embeddings and motion embeddings imply substantially different semantics, indicating unclear correspondence between texts and motions. Besides, this fashion over-compresses a sentence into one embedding, which compromises the fine-grained correspondence between each word and each motion frame. Although there are some methods (Zhang et al., 2024b; 2023b) to perform texts and motion interactions via linear cross-attention, they fuse the diffusion timestep embeddings with textual features together in the forward process. This operation undermines the structural text representations and weakens the input textual conditions. Through these observations, we argue that the fine-grained text-motion correspondence in these two motion diffusion fashions **is not well considered**. Therefore, it is urgent to build a model with good explainability and clear modeling of fine-grained text-motion correspondence.

To resolve these issues, in this work, we propose a motion diffusion model, namely MotionCLR, with a CLeaR modeling of the motion generation process and fine-grained text-motion correspondence. The main component of MotionCLR is a CLR block, which is composed of a convolution layer, a self-attention layer, a cross-attention layer, and an FFN layer. In this basic block, the cross-attention layer is used to encode the text conditions **for each word**. More specifically, the cross-attention operation between each word and each motion frame models the text-motion correspondence *explicitly*. Meanwhile, the timestep injection of the diffusion process and the text encoding are modeled separately. Besides, the self-attention layer in this block is designed for modeling the interaction between different motion frames and the FFN layer is a common design for channel mixing.

Motivated by previous progress in the explainality of the attention mechanism (Vaswani et al., 2017; Ma et al., 2023; Hao et al., 2021; Xu et al., 2015; Hertz et al., 2023; Chefer et al., 2021b;a), this work delves into the mathematical properties of the basic CLR block, especially the cross-attention and self-attention mechanisms. In the CLR block, the cross-attention value of each word along the time axis works as an activator to determine the execution time of each action. Besides, the self-attention mechanism in the CLR block mainly focuses on mining similar motion patterns between frames. Our empirical studies verify these properties. Based on these key observations, we show how we can achieve versatile motion editing downstream tasks (*e.g.* motion (de-)emphasizing, in-place motion replacement, and motion erasing) by manipulating cross-attention and self-attention calculations. We verify the effectiveness of these editing methods via both qualitative and quantitative experimental results. Additionally, we explore the potential of action counting with the self-attention map and show how our method can be applied to cope with the hallucination of generative models.

Before delving into the technical details of this work, we summarize our key contributions as follows.

- We propose an attention-based motion diffusion model, namely MotionCLR, with clear modeling of the text-aligned motion generation process. MotionCLR achieves comparable generation performance with state-of-the-art methods.
- For the first time in the human animation community, we clarify the roles that self- and cross-attention mechanisms play in one attention-based motion diffusion model.
- Thanks to these observations, we propose a series of interactive motion editing downstream tasks (see Fig. 1) via manipulating attention layer calculations. We additionally explore the potential of our method to perform grounded motion generation when facing failure cases.

## 2 RELATED WORK AND CONTRIBUTION

**Text-driven human motion generation** (Plappert et al., 2018; Ahn et al., 2018; Lin & Amer, 2018; Ahuja & Morency, 2019; Bhattacharya et al., 2021; Tevet et al., 2022a; Petrovich et al., 2022; Hong et al., 2022; Guo et al., 2022b; Zhang et al., 2024b; Athanasiou et al., 2022; Tevet et al., 2022b; Wang et al., 2022; Chen et al., 2023b; Dabral et al., 2023; Yuan et al., 2023; Zhang et al., 2023a; Shafir et al., 2024; Zhang et al., 2023b; Karunratanakul et al., 2023; Jiang et al., 2024; Zhang et al., 2024e; Xiao et al., 2024; Xie et al., 2024a; Lu et al., 2024; Wan et al., 2024; Guo et al., 2024a; Liu et al., 2024; Han et al., 2024; Xie et al., 2024b; Zhou et al., 2024; Petrovich et al., 2024; Barquero et al., 2024; Wang et al., 2024; Huang et al., 2024; Zhang et al., 2024a) uses textual descriptions as input to synthesize human motions. One of the main generative fashions is a kind of GPT-like (Zhang et al., 2023a; Lu et al., 2024; Guo et al., 2024a; Jiang et al., 2024) motion generation method, which compresses the text input into one conditional embedding and predicts motion in an auto-regressive fashion. Besides, the diffusion-based method (Tevet et al., 2022b; Zhang et al., 2024b; 2023b; Zhou et al., 2024; Chen et al., 2023b; Dai et al., 2024) is another generative fashion in motion generation. Note that most work with this fashion also utilizes transformers (Vaswani et al., 2017) as the basic network architecture. Although these previous attempts have achieved significant progress in the past years, the technical design of the explainability of the attention mechanism is still not well considered.

**Motion editing** aims to edit a motion satisfying human demand. Previous works (Dai et al., 2024; Dabral et al., 2023; Kim et al., 2023) attempt to edit a motion in a controlling fashion, like motion inbetweening and joint controlling. There are some other methods (Raab et al., 2023; Aberman et al., 2020b; Jang et al., 2022) trying to control the style of a motion. However, these works are either designed for a specific task or cannot edit fine-grained motion semantics, such as the height or times of a "jump" motion. Raab et al. (2024a) perform motion following via replacing the queries in the self-attention. Goel et al. (2024) propose to edit a motion with an instruction. However, the fine-grained text-motion correspondence in the cross-attention still lacks an in-depth understanding. There are also some methods designed for motion generation (Li et al., 2002) or editing (Lee & Shin, 1999; Holden et al., 2016; Athanasiou et al., 2024), which are limited to adapt to diverse downstream tasks. Compared to motion editing, the field of diffusion-based image editing has been largely explored. Previous studies have achieved exceptional realism and diversity in image editing (Hertz et al., 2023; Han et al., 2023; Parmar et al., 2023; Cao et al., 2023; Tumanyan et al., 2023; Zhang et al., 2023c; Mou et al., 2024; Ju et al., 2024) by manipulating attention maps. Especially, although Hertz et al. (2023) propose to introduce cross-attention into image editing, these techniques and self-attention-based motion editing are still under-explored. However, relevant interactive editing techniques and observations are still unexplored in the human animation community.

**Our key insights and contribution** over previous attention-based motion diffusion models (Tevet et al., 2022b; Zhang et al., 2024b; 2023b; Zhou et al., 2024; Chen et al., 2023b; Dai et al., 2024) lie in the clear explainability of the self-attention and cross-attention mechanisms in diffusion-based motion generation models. The cross-attention module in our method models the text-motion correspondence at the *word level* explicitly. Besides, the self-attention mechanism models the motion coherence between frames. Therefore, we can easily clarify what roles self-attention and cross-attention mechanisms play in this framework, respectively. To the best of our knowledge, it is the first time in the human animation community to clarify these mechanisms in one system and explore how to perform training-free motion editing involving humans in the loop.

## 3 BASE MOTION GENERATION MODEL AND UNDERSTANDING ATTENTION MECHANISMS

In this section, we will introduce the proposed motion diffusion model, MotionCLR, composed of several basic CLR modules. Specifically, we will analyze the technical details of the attention mechanism to obtain an in-depth understanding of this.

### 3.1 HOW DOES MOTIONCLR MODEL FINE-GRAINED CROSS-MODAL CORRESPONDENCE?

Regarding the issues of the previous methods (see Sec. 1), we carefully design a simple yet effective motion diffusion model, namely MotionCLR, with **fine-grained word-level text-motion correspondence**. The MotionCLR model is a U-Net-like architecture (Ronneberger et al., 2015). Here, we name the down/up-sampling blocks in the MotionCLR as sampling blocks. Each sampling block includes two CLR blocks and one down/up-sampling operation. In MotionCLR, the atomic block is the CLR block, which is our key design. Specifically, a CLR block is composed of four modules,

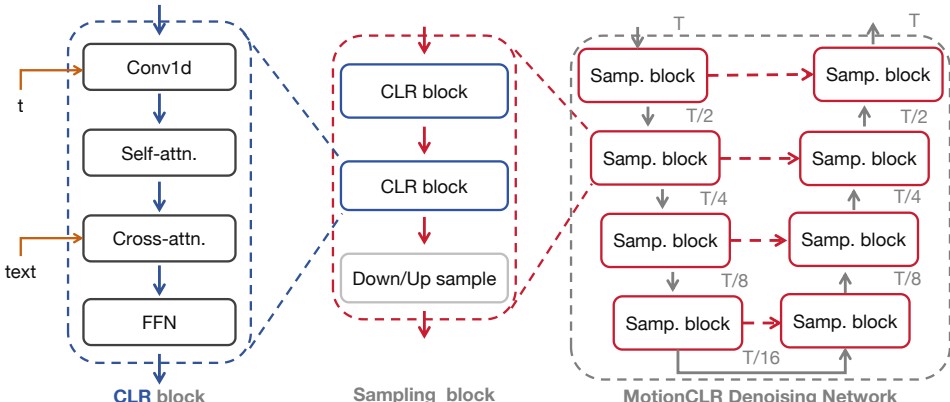

Figure 2: **System overview of MotionCLR architecture.** (a) The basic CLR block includes four layers. (b) The sampling (*a.k.a.* Samp.) block includes two CLR blocks and one down/up-sampling operation. (c) MotionCLR is a U-Net-like architecture, composed of several Sampling blocks.

- **Convolution-1D module**, *a.k.a.* Conv1d($\cdot$), is used for timestep injection, which is disentangled with the text injection. The design principle here is to disentangle the text embeddings and the timestep embeddings for explicit modeling for both conditions.

- **Self-attention module** is designed for learning temporal coherence between different motion frames. Notably, different from previous works (Tevet et al., 2022b; Zhou et al., 2024; Shafir et al., 2024), self-attention only models the correlation between motion frames and does not include any textual inputs. *The key motivation here is to separate the motion-motion interaction from the text-motion interaction of traditional fashions (Tevet et al., 2022b).*

- **Cross-attention module** plays a crucial role in learning text-motion correspondence in the CLR block. It takes word-level textual embeddings of a sentence for cross-modality interaction, aiming to obtain *fine-grained* text-motion correspondence *at the word level*. Specifically, *the attention map explicitly models the relationship between each frame and each word, enabling more fine-grained cross-modality controlling* (Detailed comparison with previous methods in Appendix C.3).

- **FFN module** works as an additional feature transformation and extraction (Dai et al., 2022; Geva et al., 2021), which is a necessary component in transformer-based architectures.

*In summary, in the basic CLR block, we model interactions between frames and cross-modal correspondence, separately and explicitly.* More detailed comparisons with previous work are in Appendix C.3. We analyze both self-attention and cross-attention of MotionCLR in following sections.

## 3.2 MATHEMATICAL PRELIMINARIES OF ATTENTION MECHANISM IN MOTIONCLR

**The general attention mechanism** has three key components, query ($\mathbf{Q}$), key ($\mathbf{K}$), and value ($\mathbf{V}$), respectively. The output $\mathbf{X}'$ of the attention mechanism can be formulated as,

$$\mathbf{X}' = \texttt{softmax}(\mathbf{Q}\mathbf{K}^\top/\sqrt{d})\mathbf{V}, \qquad (1)$$

where $\mathbf{Q} \in \mathbb{R}^{N_1 \times d}$, $\mathbf{K}, \mathbf{V} \in \mathbb{R}^{N_2 \times d}$. Here, $d$ is the embedding dimension of the text or one-frame motion. In the following section, we take $t = 0, 1, \cdots, T$ as diffusion timesteps, and $f = 1, 2, \cdots, F$ as the frame number of motion embeddings $\mathbf{X} \in \mathbb{R}^{F \times d}$. For convenience, we name $\mathbf{S} = \mathbf{Q}\mathbf{K}^\top$ as the similarity matrix and $\texttt{softmax}(\mathbf{Q}\mathbf{K}^\top/\sqrt{d})$ as the attention map in the following sections.

**The self-attention** mechanism uses different transformations of motion features $\mathbf{X}$ as inputs,

$$\mathbf{Q} = \mathbf{X}\mathbf{W}_Q, \quad \mathbf{K} = \mathbf{X}\mathbf{W}_K, \quad \mathbf{V} = \mathbf{X}\mathbf{W}_V, \qquad (2)$$

where $\mathbf{Q}, \mathbf{K}, \mathbf{V} \in \mathbb{R}^{F \times d}$, $F = N_1 = N_2$. We take a deep look at the formulation of the self-attention mechanism. As shown in Eq. (1), the attention calculation begins with a matrix multiplication operation, meaning the similarity ($\mathbf{S} = \mathbf{Q}\mathbf{K}^\top \in \mathbb{R}^{F \times F}$) between $\mathbf{Q}$ and $\mathbf{K}$. Specifically, for each row $i$ of $\mathbf{S}$, it obtains the frame most similar to frame $i$. Here $\sqrt{d}$ is a normalization term. After obtaining the similarity for all frames, the $\texttt{softmax}(\cdot)$ operation is not only a normalization function, but also works as a "soft" $\texttt{max}(\cdot)$ function for selecting the frame most similar to frame $i$. Assuming the $j$-th frame is selected as the frame most similar to frame $i$ with the maximum activation, the final multiplication with $\mathbf{V}$ will approximately replace the motion feature $\mathbf{V}_j$ at the $i$-th row of $\mathbf{X}'$. Here, the output $\mathbf{X}'$ is the updated motion feature. In summary, we have the following remark.

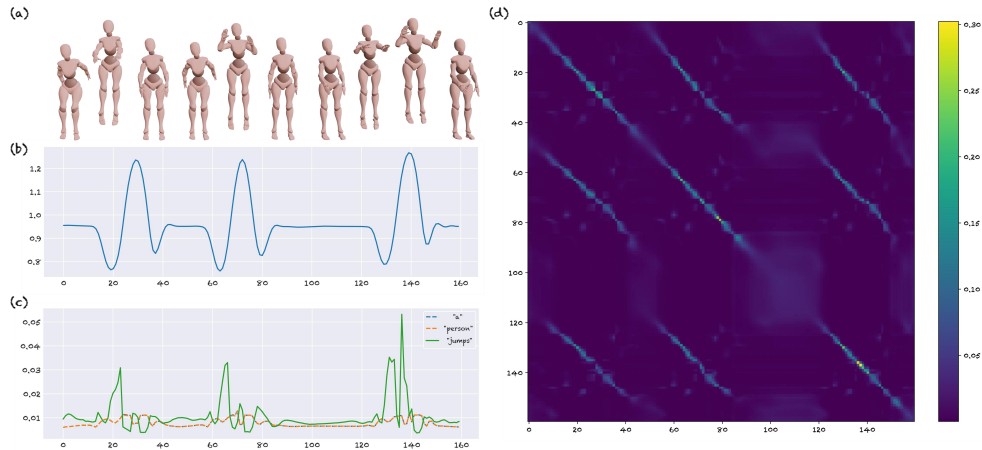

Figure 3: **Empirical study of attention mechanisms.** We use "a person jumps." as an example. (a) Key frames of generated motion. (b) The root trajectory along the $Y$-axis (vertical). The character jumps on $\sim 15 - 40\text{f}, \sim 60 - 80\text{f}$, and $\sim 125 - 145\text{f}$, respectively. (c) The **cross-attention** between timesteps and words. The "jump" word is highly activated aligning with the "jump" action. (d) The **self-attention** map visualization. It is obvious that the character jumps three times. Different jumps share similar local motion patterns.

> *Remark* 1. The self-attention mechanism measures the motion similarity of all frames and aims to select the most similar frames in motion features at each place (Detailed diagram in Appendix C).

**The cross-attention** mechanism of MotionCLR uses the transformation of a motion as a query, and the transformation of textual words as keys and values,

$$\mathbf{Q} = \mathbf{X}\mathbf{W}_Q, \quad \mathbf{K} = \mathbf{C}\mathbf{W}_K, \quad \mathbf{V} = \mathbf{C}\mathbf{W}_V, \tag{3}$$

where $\mathbf{C} \in \mathbb{R}^{L \times d}$ is the textual embeddings of $L$ word tokens, $\mathbf{Q} \in \mathbb{R}^{F \times d}$, $\mathbf{K}, \mathbf{V} \in \mathbb{R}^{L \times d}$. Note that $\mathbf{W}_\star$ in Eq. (2) and Eq. (3) are not the same parameters, but are used for convenience. As shown in Eq. (3), $\mathbf{K}$ and $\mathbf{V}$ are both the transformed text features. Recalling Eq. (1), the matrix multiplication operation between $\mathbf{Q}$ and $\mathbf{K}$ measures the similarity ($\mathbf{S} = \mathbf{Q}\mathbf{K}^\top$) between motion frames and words in a sentence. Similar to that in self-attention, the $\texttt{softmax}(\cdot)$ operation works as a "soft" $\max(\cdot)$ function to select which transformed word embedding in $\mathbf{V}$ should be selected at each frame. This operation models the motion-text correspondence explicitly. Therefore, we have the second remark.

> *Remark* 2. The cross-attention first calculates the similarity matrix to determine which word (*a.k.a.* value in attention) should be activated at the $i$-th frame explicitly. The final multiplication operation with values places the semantic features of their corresponding frames. (Detailed diagram in Appendix C)

### 3.3 Empirical Evidence on Understanding Attention Mechanisms

To obtain a deeper understanding of the attention mechanism and verify the mathematical analysis of attention mechanisms, we provide some empirical studies on some cases. Due to the page limits, we leave more visualization results for empirical evidence in Appendix D.

As shown in Fig. 3, we take the sentence "a person jumps." as an example. Besides the keyframe visualization (Fig. 3a), we also visualize the root trajectory along the $Y$-axis (vertical height, in Fig. 3b). As can be seen in Fig. 3, the character jumps at $\sim 15 - 40\text{f}, \sim 60 - 80\text{f}$, and $\sim 125 - 145\text{f}$, respectively. Note that, as shown in Fig. 3c, the word "jump" is significantly activated aligning with the "jump" action in the self-attention map. This not only verifies the soundness of the fine-grained text-motion correspondence modeling in MotionCLR, but also meets the theatrical analysis of motion-text ($\mathbf{Q}$-$\mathbf{K}$) similarity. This motivates us to manipulate the attention map to control when the action will be executed. The details will be introduced in Sec. 4.

We also visualize the self-attention map in Fig. 3d. As analyzed in Sec. 3.2, the self-attention map evaluates the similarity between frames. As can be seen in Fig. 3d, the attention map highlights **nine** areas with similar motion patterns, indicating **three** jumping actions in total. Besides the time areas that the "jmup" word is activated are aligned with the jumping actions. The highlighted areas in the self-attention map are of line shapes, indicating the taking-off, in-the-air, and landing actions of a jump with different detailed movement patterns.

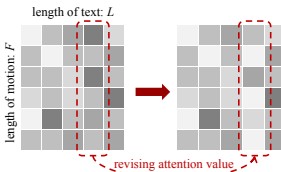 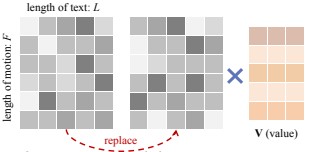 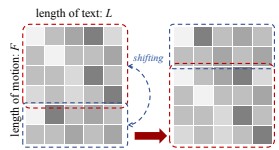

(a) Motion (de-)emphasizing via in/de-creasing cross-attention value.

(b) In-place motion replacement via replacing self-attention map.

(c) Motion sequence shifting via shifting cross-attention map.

Figure 4: **Motion editing via manipulating attention maps.**

## 4 VERSATILE APPLICATIONS VIA ATTENTION MANIPULATIONS

Analysis in Sec. 3.2 has revealed the roles that attention mechanisms play in MotionCLR. In this section, we will show versatile downstream tasks of MotionCLR via manipulating attention maps.

**Motion emphasizing and de-emphasizing.** In the text-driven motion generation framework, the process is driven by the input text. As discussed in Sec. 3.2, the verb of the action will be significantly activated in the cross-attention map when the action is executed. As shown in Fig. 4a, if we increase/decrease the attention value of a verb in the cross-attention map, the corresponding action will be emphasized/de-emphasized. Besides, this method can also be extended to the **grounded motion generation**, which will be introduced in Sec. 6.

**Motion erasing.** Motion erasing is a special case of motion de-emphasizing. We treat it as a special case of motion de-emphasizing. When the decreased (de-emphasized) cross-attention value of an action is small enough, the corresponding action will be erased.

**In-place motion replacement.** In real scenarios, we would like to edit some local motion contents of the generated result. Assuming we generate a reference motion at first, we would like to replace one action in the reference motion with another in place. Therefore, the batch size of inference examples is two during the inference stage, where the first is the reference motion and the other is the edited motion. As discussed in Sec. 3.2, the cross-attention map determines when an action happens. Motivated by this, we replace the cross-attention map of the edited motion as the one of the reference motion. As shown in Fig. 4b, we use the replaced attention map to multiply the value matrix (text features) to obtain the output.

**Motion sequence shifting.** It is obvious that the generated motion is a combination of different actions along the time axis. Sometimes, users would like to shift a part of the motion along the time axis to satisfy the customized requirements. As shown in Fig. 4c, we can shift the motion sequentiality by shifting the self-attention map. As discussed in Sec. 3.2, self-attention is only related to the motion feature without related to the semantic condition, which is our motivation on manipulating the self-attention map. Thanks to the denoising process, the final output sequence should be a natural and continuous sequence.

**Example-based motion generation.** As defined by Li et al. (2023b), example-based motion generation aims to generate novel motions referring to an example motion. In MotionCLR system, this task is a special case of the motion sequence shifting. That is to say, we can shuffle the queries of the self-attention map to obtain the diverse motions referring to the example.

**Motion style transfer.** As discussed in the technical details of the self-attention mechanism, the values mainly contribute to the contents of motion and the attention map determines the selected indices of motion frames. When synthesizing two motion sequences ($\mathbf{M}_1$ and $\mathbf{M}_2$ respectively), we only need to replace $\mathbf{Q}$s in $\mathbf{M}_2$ with that in $\mathbf{M}_1$ to achieve the style of $\mathbf{M}_2$ into $\mathbf{M}_1$'s. Specifically, queries ($\mathbf{Q}$s) in $\mathbf{M}_2$ determine which motion feature in $\mathbf{M}_2$ is the most similar to that in $\mathbf{M}_1$ at each timestep. Accordingly, these most similar motion features are selected to compose the edited motion. Besides, the edited motion is with the motion content of $\mathbf{M}_2$ while imitating the motion style of $\mathbf{M}_1$.

We leave more technical details and pseudo codes in Appendix F.

## 5 EXPERIMENTS

### 5.1 MOTIONCLR MODEL EVALUATION

The implementations of the MotionCLR are in Appendix E.1. We first evaluate the generation performance of the MotionCLR. We extend the evaluation metrics of previous works (Guo et al., 2022a). **(1) Motion quality:** FID is adopted as a metric to evaluate the distributions between the generated and real motions. **(2) Motion diversity:** MultiModality (MModality) evaluates the diversity

| Methods | R-Precision↑ | | | FID↓ | MM-Dist↓ | Multi-Modality↑ |
|---|---|---|---|---|---|---|
| | Top 1 | Top 2 | Top 3 | | | |
| TM2T (2022b) | $0.424^{\pm0.003}$ | $0.618^{\pm0.003}$ | $0.729^{\pm0.002}$ | $1.501^{\pm0.017}$ | $3.467^{\pm0.011}$ | $2.424^{\pm0.093}$ |
| T2M (2022a) | $0.455^{\pm0.003}$ | $0.636^{\pm0.003}$ | $0.736^{\pm0.002}$ | $1.087^{\pm0.021}$ | $3.347^{\pm0.008}$ | $2.219^{\pm0.074}$ |
| MDM (2022b) | - | - | $0.611^{\pm0.007}$ | $0.544^{\pm0.044}$ | $5.566^{\pm0.027}$ | $\mathbf{2.799}^{\pm0.072}$ |
| MLD (2023b) | $0.481^{\pm0.003}$ | $0.673^{\pm0.003}$ | $0.772^{\pm0.002}$ | $0.473^{\pm0.013}$ | $3.196^{\pm0.010}$ | $2.413^{\pm0.079}$ |
| MotionDiffuse (2024b) | $0.491^{\pm0.001}$ | $0.681^{\pm0.001}$ | $0.782^{\pm0.001}$ | $0.630^{\pm0.001}$ | $3.113^{\pm0.001}$ | $1.553^{\pm0.042}$ |
| T2M-GPT (2023a) | $0.492^{\pm0.003}$ | $0.679^{\pm0.002}$ | $0.775^{\pm0.002}$ | $0.141^{\pm0.005}$ | $3.121^{\pm0.009}$ | $1.831^{\pm0.048}$ |
| ReMoDiffuse (2023b) | $0.510^{\pm0.005}$ | $0.698^{\pm0.006}$ | $0.795^{\pm0.004}$ | $0.103^{\pm0.004}$ | $2.974^{\pm0.016}$ | $1.795^{\pm0.043}$ |
| MoMask (2024a) | $0.521^{\pm0.002}$ | $0.713^{\pm0.002}$ | $0.807^{\pm0.002}$ | $\mathbf{0.045}^{\pm0.002}$ | $2.958^{\pm0.008}$ | $1.241^{\pm0.040}$ |
| MotionCLR | $0.542^{\pm0.001}$ | $\mathbf{0.733}^{\pm0.002}$ | $0.827^{\pm0.003}$ | $0.099^{\pm0.003}$ | $2.981^{\pm0.011}$ | $2.145^{\pm0.043}$ |
| MotionCLR* | $\mathbf{0.544}^{\pm0.001}$ | $0.732^{\pm0.001}$ | $\mathbf{0.831}^{\pm0.002}$ | $0.269^{\pm0.001}$ | $\mathbf{2.806}^{\pm0.014}$ | $1.985^{\pm0.044}$ |

Table 1: **Comparison with different methods on the HumanML3D dataset.** The "*" notation denotes the DDIM sampling inference design choice and the other is the DPM-solver sampling choice.

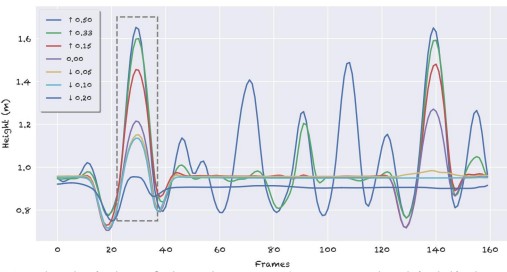

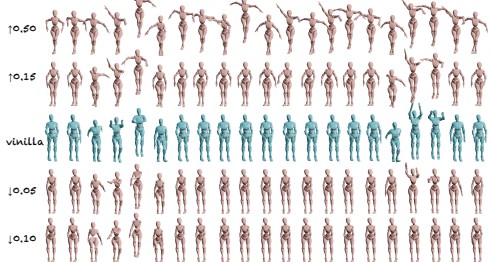

(a) The height of the character's root. The highlighted area is obvious when comparing different weights.

(b) Visualization of the edited motions on different (de-)emphasizing weight settings.

Figure 5: **Motion (de-)emphasizing.** Different weights of "jump" (↑ or ↓) in "a man jumps.".

based on the same text and diversity calculates variance among features. **(3) Text-motion matching:** Following Guo et al. (2022a), we calculate the R-Precision to evaluate the text-motion matching accuracy and MM-Dist to show the distance between texts and motions. The results are shown in Tab. 1, indicating a comparable performance with the state-of-the-art method. Especially, our result has a higher text-motion alignment over baselines, owing to the explicit fine-grained cross-modality modeling. As shown in Tab. 1, both DDIM and DPM-solver sampling work consistently well compared with baselines. We leave more visualization and qualitative results in Appendix A.

## 5.2 INFERENCE ONLY MOTION EDITING

**Motion (de-)emphasizing and motion erasing.** For quantitative analysis, we construct a set of prompts to synthesize motions, annotating the key verbs in the sentence by human researchers. The metric here is the TMR similarity (TMR-sim.) (Petrovich et al., 2023) used for measuring the text-motion similarity (between 0 and 1, with % in table). The comparison in Tab. 2 shows the de-emphasizing makes the motion less similar to text, and emphasizing ones are more aligned at the beginning of increasing weights. When weights are too large, the attention maps are corrupted, resulting in artifacts. Therefore, the suggested value

| weight | TMR-sim. (%) | FID |
|---|---|---|
| - 0.60 | 52.059 | 0.776 |
| - 0.50 | 52.411 | 0.394 |
| - 0.40 | 53.294 | 0.235 |
| - 0.30 | 53.364 | 0.225 |
| baseline | 53.956 | 0.217 |
| + 0.30 | 54.311 | 0.210 |
| + 0.40 | 54.496 | 0.208 |
| + 0.50 | 54.532 | 0.223 |
| + 0.60 | 54.399 | 0.648 |

Table 2: Abaltion on motion (de-)emphasizing.

of the weights ranges from $-0.5$ to $+0.5$. We mainly provide the visualization results of motion (de-)emphasizing in Fig. 5. As shown in Fig. 5, the edited results are aligned with the manipulated attention weights. Especially, as can be seen, in Fig. 5a, the height of the "jump" action is accurately controlled by the cross-attention weight of the word "jump". For an extremely large adjusting weight, *e.g.* ↑1.0, the times of the jumping action also increase. This is because the low-activated timesteps of the vanilla generated motion might have a larger cross-attention value to activate the "jump" action. As motion erasing is a special case of motion de-emphasizing, we do not provide more quantitative on this application. We provide some visualization results in Fig. 6. As can be seen in Fig. 6a, the second jumping action is erased. Besides, the "waving hand" case shown in Fig. 6b shows that the final 1/3 waving action is also removed. More experiments are in Appendix A.1 and A.2.

**In-place motion replacement.** Different from naïve replacing prompts for motion replacement, in-place motion replacement not only replaces the original motion at the semantic level, but also needs to replace motions at the exact temporal place. Fig. 7a and Fig. 7b show the root height trajectory and the root horizontal velocity, respectively. In this case, the edited and original motion share the same time zone to execute the action. Besides, the edited motion is semantically aligned with the "walk". Fig. 7c also shows results of replacing "runs" as "jumps" without changing the sitting action.

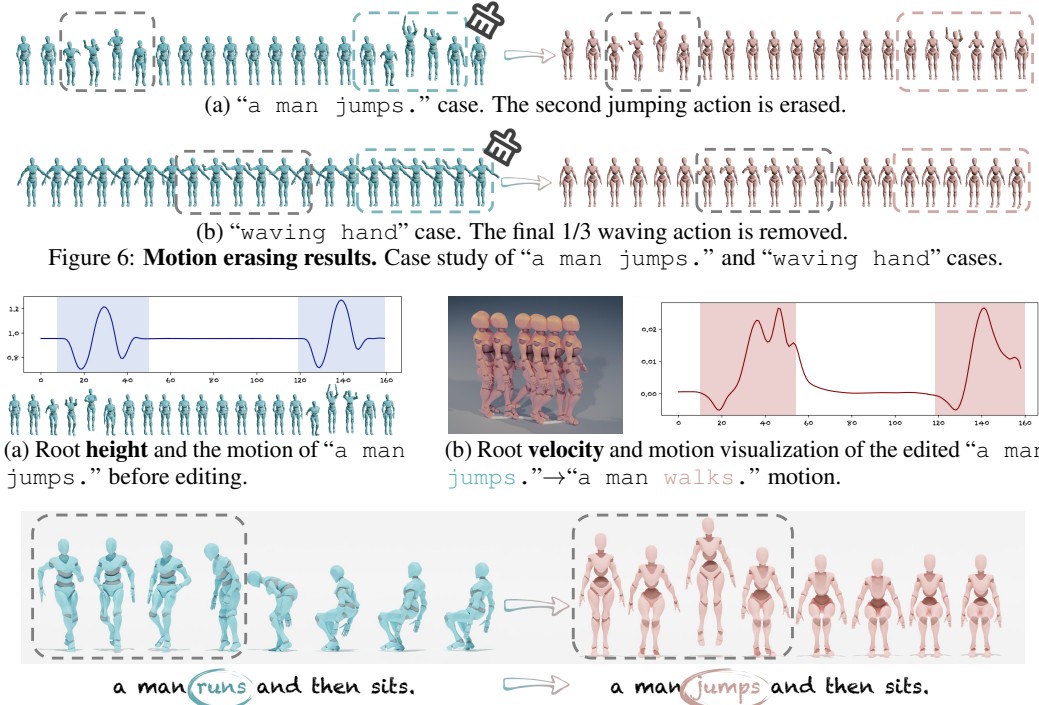

(a) "`a man jumps.`" case. The second jumping action is erased.

(b) "`waving hand`" case. The final 1/3 waving action is removed.

Figure 6: **Motion erasing results.** Case study of "`a man jumps.`" and "`waving hand`" cases.

(a) Root **height** and the motion of "`a man jumps.`" before editing.

(b) Root **velocity** and motion visualization of the edited "`a man jumps.`"→"`a man walks.`" motion.

(c) Motion in-place replacement results of a motion including multiple actions.

Figure 7: **In-place motion replacement.** Replacing the "`jumps`" in "`a man jumps.`" as "`walks`", the edited motion and the vanilla motion share the same temporal area of the action execution.

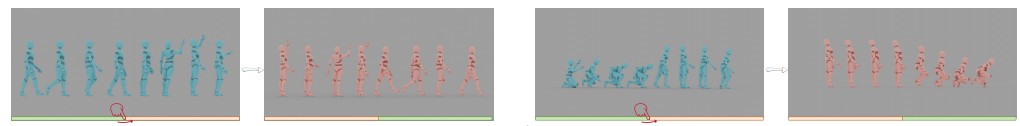

(a) Prompt: "`a person walks straight and then waves.`" Original (blue) *vs.* shifted (red) motion.

(b) Prompt: "`a man gets up from the ground.`" Original (blue) *vs.* shifted (red) motion.

Figure 9: **Comparison between original and shifted motions.** Time bars are shown in different colors. (a) The original figure raises hands after the walking action. The shifted one has the opposite sequentiality. (b) The key squatting action is shifted to the end of the sequence, and the standing-by action is shifted to the beginning.

**Motion sequence shifting.** Here, we provide some comparisons between the original motion and the edited one. In Fig. 9, we take "▭" and "▭" to represent different time bars, whose orders represent the sequentially. As can be seen in Fig. 9a, the execution of waving hands is shifted to the beginning of the motion. Besides, as shown in Fig. 9b, the squatting action has been moved to the end of the motion. These results show that the editing of the self-attention map sequentiality has an explicit correspondence with the editing motion sequentially. More results are in Appendix A.6.

**Example-based motion generation.** The example-based motion generation (Li et al., 2023b) task has two basic requirements. (1) *The first one is the generated motions should share similar motion textures (Li et al., 2002) with the example motion.* We observe the high-dimension structure of motions via dimensionality reduction. As the t-SNE visualization results shown in Fig. 8, generated motions driven by the same example are similar to the given example (in the same color). (2) *Besides, different generated motions driven by the same example should be diverse.* As shown in Fig. 10, these generated results are diverse not only in local motions (Fig. 10a) but also in the global trajectory (Fig. 10b). Furthermore, results in Fig. 10 also share the similar motion textures. We leave more visualization results in Appendix A.7.

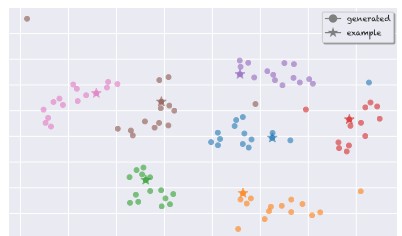

Figure 8: **t-SNE visualization of different example-based generated results.** Different colors imply different driven examples.

**Motion style transfer.** As shown in Fig. 11, in the MotionCLR framework, the style reference motion provides style and the content reference motion provides keys and values. As can be seen in Fig. 11, all edited results are well-stylized with style motions and keep the main movement content with the content reference.

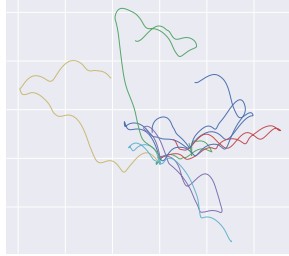

| | |
|---|---|
| (a) Examples (blue) and generated (red) motions. | (b) Root trajectory visualization. |

Figure 10: **Diverse generated motions driven by the same example.** Prompt: "`a person steps sideways to the left and then sideways to the right.`". (a) The diverse generated motions driven by the same example motion share similar movement content. (b) The root trajectories of diverse motions are with similar global trajectories, but not the same.

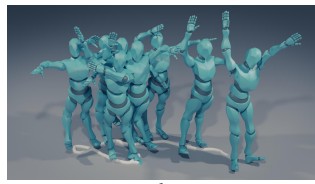 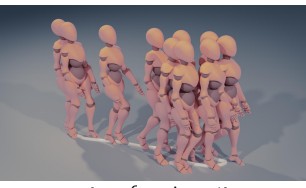

style reference          content reference          transferred result

(a) Style reference: "`the person dances very happily`", content reference: "`the man is walking`". The transferred result shows a figure walking in a back-and-forth happy pace.

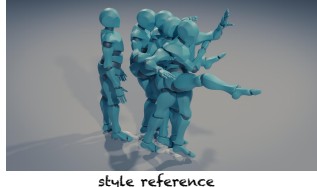 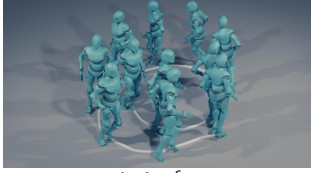 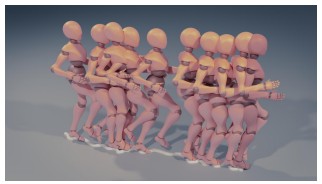

style reference          content reference          transferred result

(b) Style reference: `a man is doing hip-hop dance`", Content reference: `a person runs around a circle`". The stylized result shows a running motion with bent hands, shaking left and right.

Figure 11: **Motion style transfer results.** The style reference, content references, and the transferred results are shown from left to right for each case.

## 5.3 ABLATION STUDY

We provide some ablation studies on some technical designs. (1) The setting *w/o separate word modeling* shows poorer qualitative results with the w/ separate word setting. The separate word-level cross-attention correspondence benefits better text-to-motion controlling, which is critical for motion fine-grained generation. (2) The setting of *injecting text tokens before motion tokens* performs worse than the MotionCLR. This validates the effectiveness of introducing the cross-attention for cross-modal correspondence. The ablation studies additionally verify the basic motivation of modeling word-level correspondence in MotionCLR.

| Ablation | R-Precision↑ | | | FID↓ |
|---|---|---|---|---|
| | Top 1 | Top 2 | Top 3 | |
| (1) | 0.512 | 0.705 | 0.792 | 0.544 |
| (2) | 0.509 | 0.703 | 0.788 | 0.550 |
| MotionCLR | **0.544** | **0.732** | **0.831** | **0.269** |

Table 3: Ablation studies between different technical design choices.

## 5.4 ACTION COUNTING FROM ATTENTION MAP

As shown in Fig. 3, the number of executed actions in a generated motion sequence can be accurately calculated via the self-attention map. We directly detect the number of peaks in each row of the self-attention map and finally average this of each row. In the technical implementation, to avoid sudden peaks from being detected, we apply average downsampling and Gaussian smoothing (parameterized by standard deviation $\sigma$). We leave more technical details in Appendix G.

We construct a set of text prompts corresponding to different actions to perform the counting capability via the self-attention map. The counting number of actions is

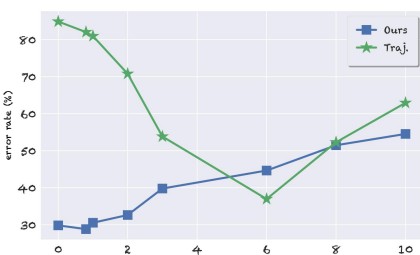

Figure 12: Action counting error rate comparison. Root trajectory (Traj.) *vs.* attention map (Ours). "$\sigma$" is the smoothing parameter.

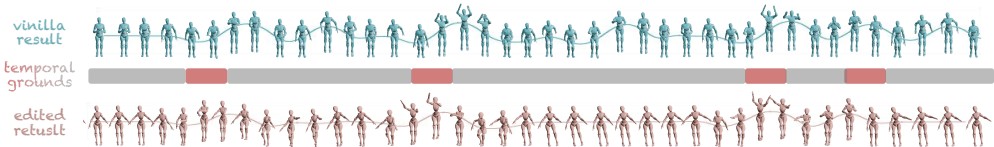

Figure 13: **Comparison between w/ *vs.* w/o grounded motion generation settings.** The root height and motion visualization of the textual prompt "a person jumps four times".

labeled by professional researchers. The details of the evaluation set are detailed in Appendix E.3. As the "walking" action is composed of sub-actions of two legs, the atomic unit of this action counting is set as 0.5. We compare our method to counting with the vertical root trajectory (Traj.) peak detection. As shown in Fig. 12, counting with the self-attention map mostly works better than counting with root trajectory. Both settings use Gaussian smoothing to blur some jitters. Our method does not require too much smoothing regularization due to the smoothness of the attention map, while counting with root trajectory needs this operation. This case study reveals the effectiveness of understanding the self-attention map in MotionCLR.

## 6 FAILURE CASES ANALYSIS AND CORRECTION

There are few generative methods that can escape the curse of hallucination. In this section, we will discuss some failure cases of our method and analyze how we can refine these results. The hallucination of counting is a notoriously tricky problem for generative models, attracting significant attention in the community and lacking a unified technical solution. Considering that this problem cannot be thoroughly resolved, we try to partially reveal this issue by additionally providing temporal grounds. For example, if the counting number of an action is not aligned with the textual prompt, we can correct this by specifying the temporal grounds of actions. Technically, the temporal mask can be treated as a sequence of weights to perform the motion emphasizing and de-emphasizing. Therefore, grounded motion generation can be easily achieved by adjusting the weights of words.

Specifically, we show some failure cases of our method. As shown in Fig. 13, the generated result of "a person jumps four times" fails to show *four* times of jumping actions, but *seven* times. To meet the requirement of counting numbers in the textual prompts, we additionally input a temporal mask, including *four* jumping timesteps, to provide temporal grounds. From the root height visualization and the motion visualization, the times of the jumping action have been successfully corrected from *seven* to *four*. Therefore, our method is promising for *grounded motion generation* to reveal the hallucination of deep models.

Moreover, other editing fashions are also potential ways to correct hallucinations of generated results. For example, the motion sequence shifting and in-place motion replacement functions can be used for correcting sequential errors and semantic misalignments, respectively.

## 7 CONCLUSION AND FUTURE WORK

**Conclusion.** In this work, we propose a diffusion-based motion generation model, MotionCLR. With this model, we carefully clarify the self-attention and the cross-attention mechanisms in the MotionCLR. Based on both theoretical and empirical analysis of the attention mechanisms in MotionCLR, we developed versatile motion editing methods. Additionally, we not only verify the action counting ability of attention maps, but also show the potential of motion corrections. We build a **user interface** in Appendix B to demonstrate how can our method support interactive editing.

**Limitation and future work.** As shown in Sec. 6, our model can also not escape the hallucination curse of generative models. Therefore, we leave the grounded motion generation as future work. As discussed in Lu et al. (2024), the CLIP model used in MotionCLR is still a bit unsatisfactory. Therefore, we will provide token-level text-motion alignment encoders to provide textual conditions. Similar to other generative models, our method also meets some issues on some extreme and OOD examples, which will be resolved by our future scalable generation solution.

**Broader impact statement.** The development of MotionCLR, a diffusion-based motion generation model, has the potential to impact various fields of motion synthesis and editing significantly. However, the complexity of the MotionCLR and its performance limitations under certain conditions may lead to errors or inaccuracies in practical applications. This could result in negative consequences in critical fields such as humanoid simulation and autonomous driving. Therefore, it is necessary to further optimize the model and carefully assess its reliability before widespread deployment.

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

APPENDIX

CONTENTS

# A  SUPPLEMENTAL EXPERIMENTS

## A.1  WHAT IS THE SELF-ATTENTION MAP LIKE IN MOTION (DE-)EMPHASIZING?

This experiment is an extension of the experiment shown inm Fig. 5.

We provide more examples of how increasing or decreasing weights impact motion (de-)emphasizing and erasing. As seen in Fig. 14, the attention maps illustrate that reducing the weights (*e.g.*, ↓ 0.05 and ↓ 0.10) results in less activations, while increasing weights (*e.g.*, ↑ 0.33 and ↑ 1.00) leads to more activations. The vanilla map serves as a reference without any adjustments. However, as indicated, excessively high weights such as ↑ 1.00 introduce some artifacts, emphasizing the need for careful tuning of weights to maintain the integrity of the generated motion outputs. This demonstrates the importance of careful weight tuning to achieve the desired motion emphasis or erasure.

Compared to Fig. 14a, Fig. 14b shows two fewer trajectories. This reduction is due to the de-emphasizing effect, where the character's second jump was not fully executed, resulting in just an arm motion (Fig. 5b). Consequently, the two actions became distinguishable, leading to fewer detected two trajectories. In Fig. 14c, the second jumping has been completely erased, resulting in only one trajectory, further demonstrating how de-emphasizing significantly affects motion execution.

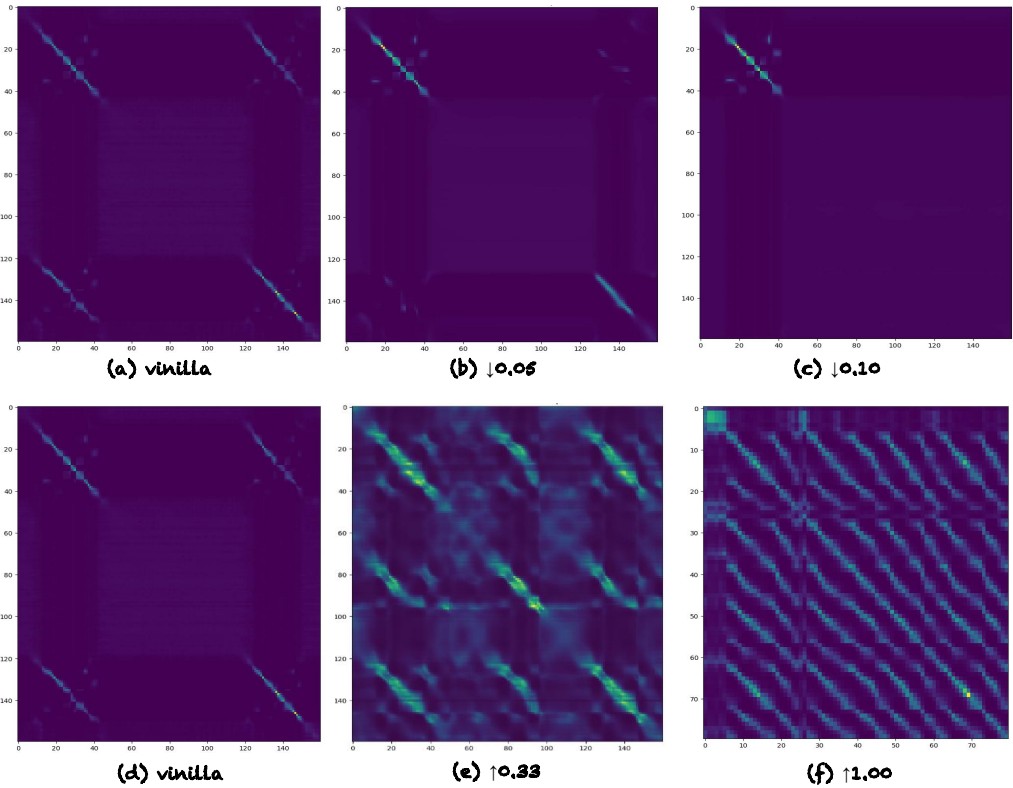

Figure 14: **Additional visualization results for different (de-)emphasizing weights.** The self-attention maps show how varying the different weights (*e.g.*, ↓ 0.05, ↓ 0.10, ↑ 0.33, and ↑ 1.00) affect the emphasis on motion.

## A.2 WHAT IS THE DIFFERENCE BETWEEN MOTION (DE-)EMPHASIZING IN MOTIONCLR AND ADJUSTING CLASSIFIER-FREE GUIDANCE WEIGHTS?

In this part, we would like to discuss the difference between reweighting the cross-attention map and adjusting classifier-free guidance weights.

As shown in Tab. 4, we ablate how different $w$s affect the results. The results suggest that adjustment of $w$ impacts the quality of generated results, making $w = 2.5$ an effective choice. When $w$ increases, the text-motion alignment increases consistently, and the generation quality (FID) requires a trade-off.

| $w$ | 1 | 1.5 | 2 | 2.5 | 3 | 3.5 |
|---|---|---|---|---|---|---|
| FID | 0.801 | 0.408 | 0.318 | 0.217 | 0.317 | 0.396 |
| TMR-sim. (%) | 51.987 | 52.351 | 53.512 | 53.956 | 54.300 | 54.529 |

Table 4: **Different editing results when changing $w$s.** In MotionCLR, $w = 2.5$ is the default design choice for the denoising sampling. All TMR-sim. metrics are timed by 100.

**However**, as the classifier-free guidance mainly works for the semantic alignment between text and motion, it cannot control the weight of each word. We take the "a man jumps." as an example for a fair comparison, which is the case used in the main text[1]. As shown in Fig. 15, the generated motions with different $w$ values illustrate that $w$ **cannot** influences both the height and frequency of the jump. Nevertheless, the classifier-free guidance is limited in its ability to control more detailed aspects, such as the exact height and number of actions. Therefore, while $w$ improves text-motion alignment, it cannot achieve fine-grained adjustments.

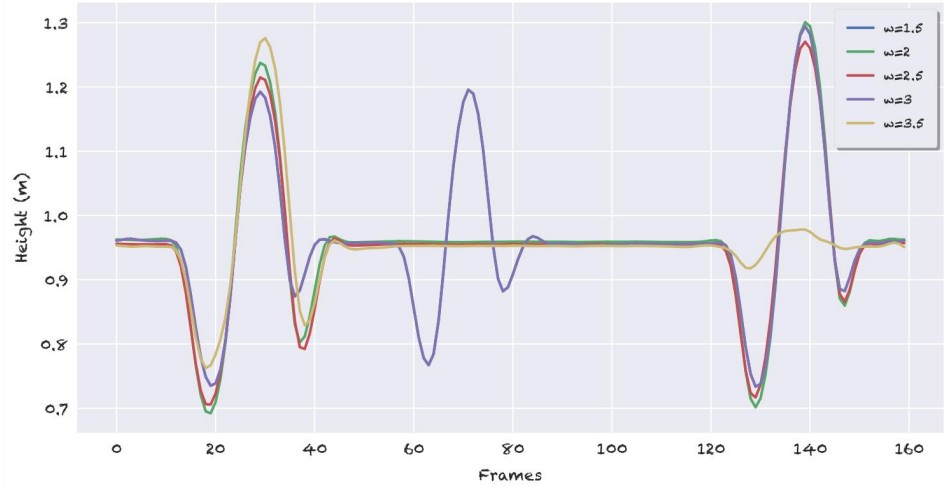

Figure 15: **The effect of varying $w$ in classifier-free guidance on generated motions.** While changing $w$ influences the general alignment between the text "a man jumps." and the generated motion, it does not provide precise control over finer details like jump height and frequency.

---

[1] Suggested to refer to Fig. 5 for comparison.

### A.3 MORE EXPERIMENTAL RESULTS OF IN-PLACE MOTION REPLACEMENT

**Semantic similarity of edited motions.** In the in-place motion replacement application, we measure the editing quality and the text-motion similarity. To verify this, we construct a set of prompts, tagged with the edited words. In Tab. 5, we compare our method (ours replaced) with the unedited ones (unreplaced) and the generated motions directly changing prompts (pseudo-GT). As can be seen in Tab. 5, the motions of the three groups have similar qualities. Besides, the edited motion is similar to the pseudo-GT group, indicating the good semantic alignment of the edited results.

| | FID $\downarrow$ | TMR-sim.$\rightarrow$ |
|---|---|---|
| direct (pseudo GT) | 0.315 | 0.543 |
| unreplaced | 0.325 | 0.567 |
| unreplaced (unpaired T-M) | 0.925 | 0.490 |
| ours replaced | 0.330 | 0.535 |

Table 5: **Comparison between the generation result with directly changing prompts and the in-place replacement in MotionCLR.** The semantics of editing results are similar to the motion directly generated by the changed prompt. The setting difference between "unreplaced" with "unreplaced (unpaired T-M)" is that the latter texts are edited sentences. All TMR-sim. are not multiplied by 100.

**Ablation study of different attention layers.** To further explore the impact of attention manipulation in in-place motion replacement, we conduct an ablation study by varying the layers in MotionCLR for manipulation, shown in Tab. 6. The table lists the results for different ranges of manipulated attention layers. It can be observed that manipulating different attention layers influences the editing quality and the semantic similarity (TMR-sim.). In particular, manipulating the layers from 1 to 18 achieves the best semantic consistency, demonstrating the effectiveness of editing across multiple attention layers for maintaining semantic alignment in the edited motion. The less effectiveness of manipulating middle layers is mainly due to the fuzzy semantics present in the middle layers of the U-Net. As these layers capture more abstract with reduced temporal resolution, the precise details and localized information become less distinct. Consequently, manipulating these layers has a limited impact on the final output, as they contribute less directly to the fine-grained details for the task.

| begin | end | FID$\downarrow$ | TMR-sim.$\uparrow$ |
|---|---|---|---|
| 8 | 11 | 0.339 | 0.472 |
| 5 | 14 | 0.325 | 0.498 |
| 1 | 18 | 0.330 | 0.535 |

Table 6: **The ablation study of manipulating different attention layers.** The "begin" and "end" represent the beginning and the final layer for manipulation. All TMR-sim. are not multiplied by 100.

## A.4 COMPARISON WITH MANIPULATION NOISY MOTIONS IN THE DIFFUSION PROCESS

As the diffusion denoising process can manipulate the motion directly in the denoising process, this is a baseline for comparison with our motion shifting and example-based motion generation applications. Here, for convenience, we only take the example-based motion generation application as an example for discussion. In this section, we conduct a comparison between our proposed editing method and diffusion manipulation in the motion space, focusing on the FID and diversity metrics. The 200 samples used in this experiment were constructed by researchers. As depicted in Tab. 7, the "Diff. manipulation" serves for our comparison. Our method achieves an FID value of 0.427, indicating a relatively high generation quality, while the "Diff. manipulation" achieves a higher FID of 0.718, demonstrating worse fidelity. Conversely, in terms of diversity, the "MotionCLR manipulation" exhibits a higher diversity (Div.) score of 2.567 compared to the 1.502 of the "Diff. manipulation." These results verify our method is better than manipulating noisy motions in the denoising process. The main reason for the better quality and diversity mainly relies on the many times of manipulation of self-attention, but not the motion. Directly manipulating the motion results in some jitters, making more effort for models to smooth. Besides, the shuffling times of manipulating the self-attention maps are higher than the baseline, contributing to the better diversity of our method.

|  | FID ↓ | Div. ↑ |
| --- | --- | --- |
| Diff. manipulation | 0.718 | 1.502 |
| MotionCLR manipulation | **0.427** | **2.567** |

Table 7: **Comparison on FID and diversity values with manipulating self-attention in the motion space of the denoising process.**

## A.5 MOTION GENERATION RESULT VISUALIZATION

We randomly chose some examples of the motion generation results in Fig. 16. The visualization results demonstrate that MotionCLR can generate coherent and realistic human motions based on diverse textual descriptions. The generated sequences capture various actions ranging from simple gestures to more complex movements, indicating the capability to handle a wide range of human behaviors. Overall, the qualitative results suggest that MotionCLR effectively translates textual prompts into human-like motions with a clear understanding of texts. This demonstrates the potential for applications in scenarios requiring accurate motion generation based on natural language inputs.

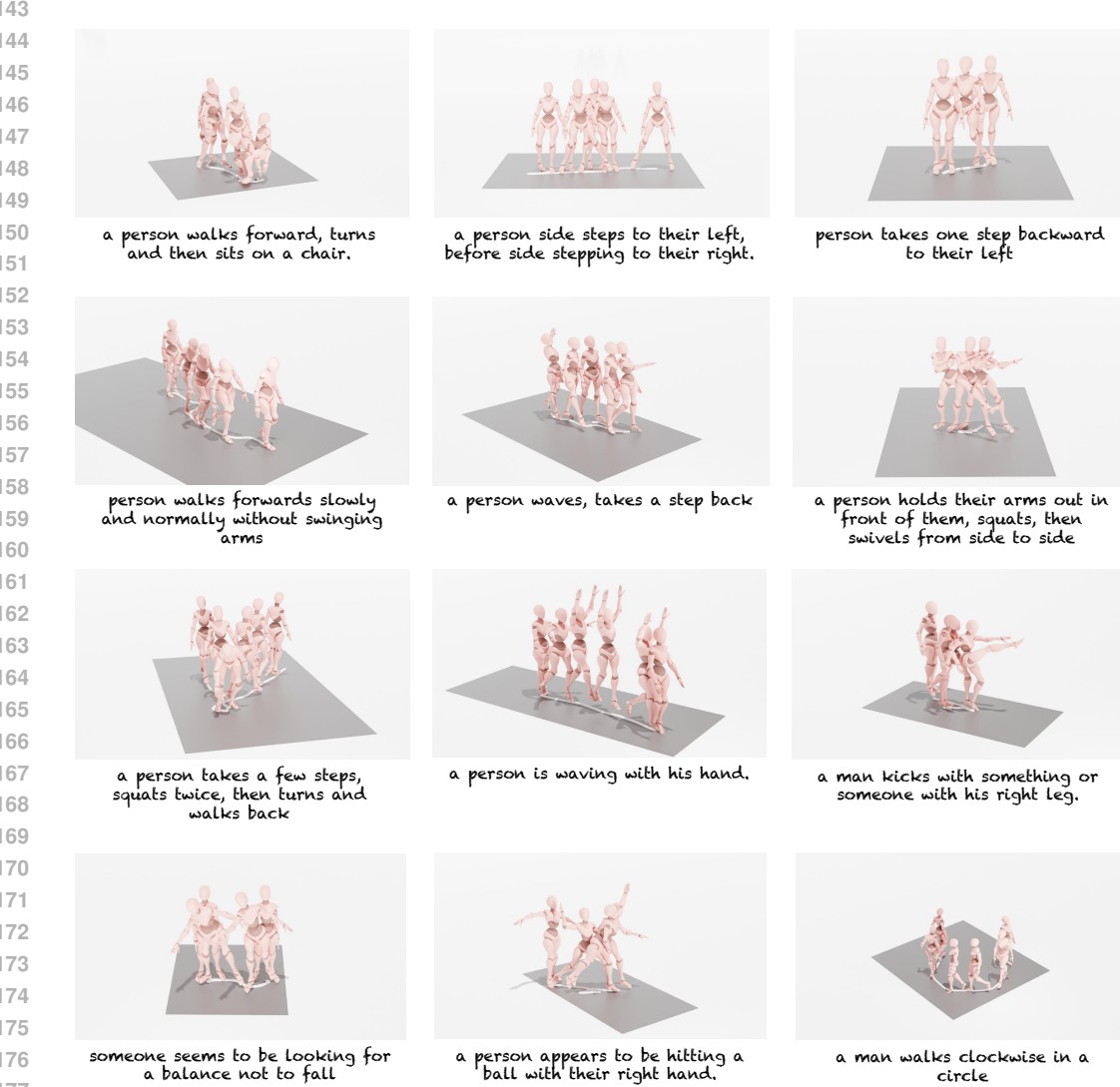

Figure 16: **Human motion generation result of MotionCLR.**

### A.6 MORE VISUALIZATION RESULTS OF MOTION SEQUENCE SHIFTING

We present further comparisons between the original and edited motions in Fig. 17. The time bars, indicated by "▭" and "▭," represent distinct phases of the motion, with their sequential arrangement reflecting the progression of the motion over time.

In Fig. 17a, we observe that the action of crossing the obstacle, originally positioned earlier in the sequence, is shifted towards the end in the edited version. This adjustment demonstrates the model's capacity to rearrange complex motions effectively while maintaining coherence. Similarly, Fig. 17b shows the standing-by action being relocated to the end of the motion sequence. This change emphasizes the model's ability to handle significant alterations in the temporal arrangement of actions. These results collectively indicate that our editing process, driven by the attention map sequentiality, exhibits a high level of correspondence with the intended edits to the motion's sequence. The model accurately captures and replicates the desired modifications, ensuring that the restructured motion retains a natural and logical flow, thereby validating the effectiveness of our motion editing approach.

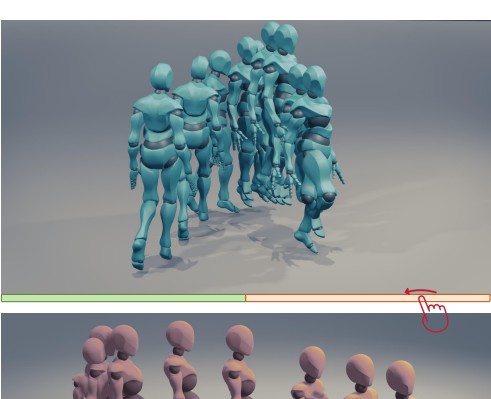
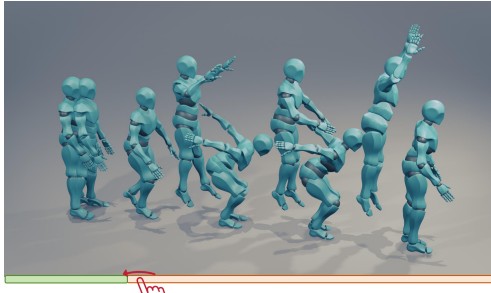
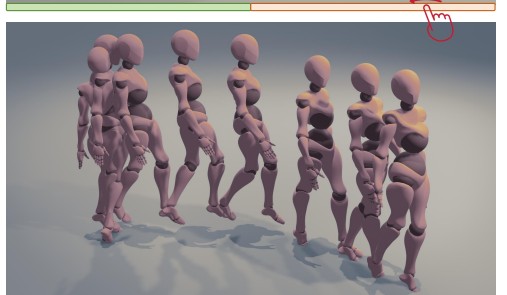
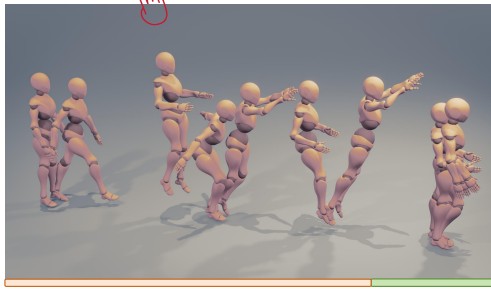

(a) Prompt: " the person is walking forward on uneven terrain." Original (blue) *vs.* shifted (red) motion.

(b) Prompt: "a person walks then jumps." Original (blue) *vs.* shifted (red) motion.

Figure 17: **Comparison between original motion and the shifted motion.** The shifted time bars are shown in different colors. (a) The original figure crosses the obstacle after the walking action. The shifted motion has the opposite sequentiality. (b) The key walking and jumping actions are shifted to the beginning of the sequence, and the standing-by action is shifted to the end.

## A.7 MORE VISUALIZATION RESULTS ON EXAMPEL-BASED MOTION GENERATION

We provide some visualization results to further illustrate the effectiveness of our approach in generating diverse motions that adhere closely to the given prompts. In Fig. 18, the example motion of "a person kicking their feet" is taken as the reference, and multiple diverse kick motions are generated. These generated motions not only exhibit variety but also maintain key characteristics of the original example. Similarly, in Fig. 19, the example motion of "a person walking in a semi-circular shape while swinging arms slightly" demonstrates the capability to generate diverse walking motions that maintain the distinct features of the source motion. The generated trajectories, as visualized in Fig. 18b and Fig. 19b, show that the diverse motions follow different paths while retaining similarities with the original motion, confirming the effectiveness of our method.

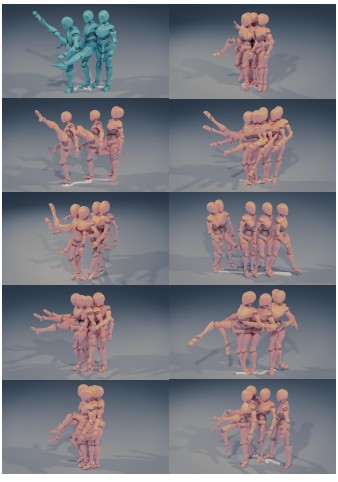
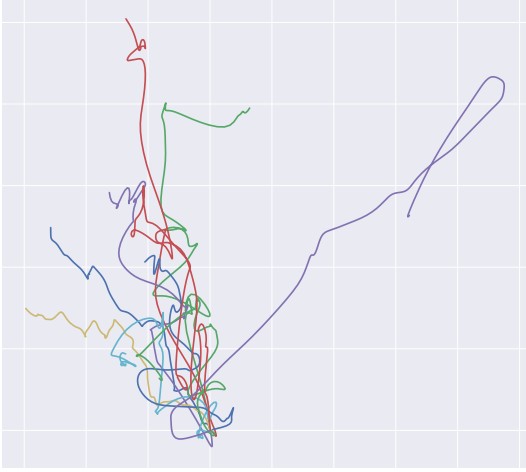

(a) The example motion (blue) and the generated diverse motion (red).

(b) The trajectory visualizations of the example motion and diverse motions.

Figure 18: **Diverse generated results of blue example generated by the prompt "a person kicks their feet.".** The example-based generated kick motions are diverse and similar to the source example.

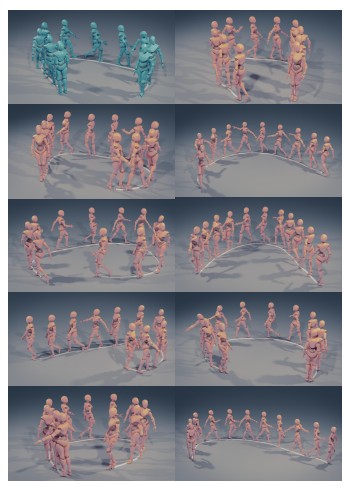
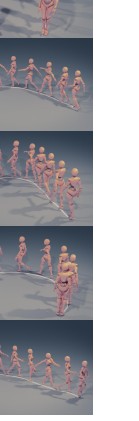
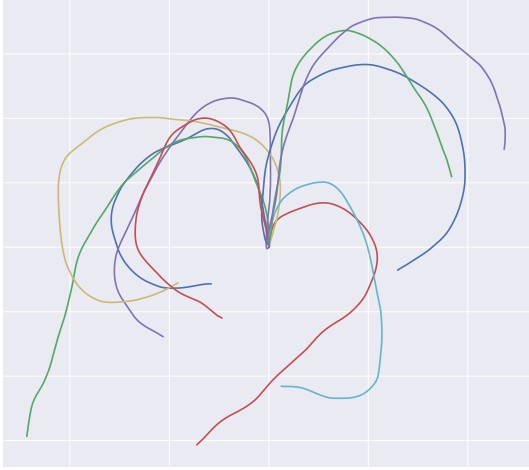

(a) The example motion (blue) and the generated diverse motion (red).

(b) The trajectory visualizations of the example motion and diverse motions.

Figure 19: **Diverse generated results of blue example generated by the prompt "person walks in a semi circular shape while swinging arms slightly.".** The example-based generated walking motions are diverse and similar to the source walking example.

## A.8 DETAILED VISUALIZATION RESULTS OF GROUNDED MOTION GENRATION

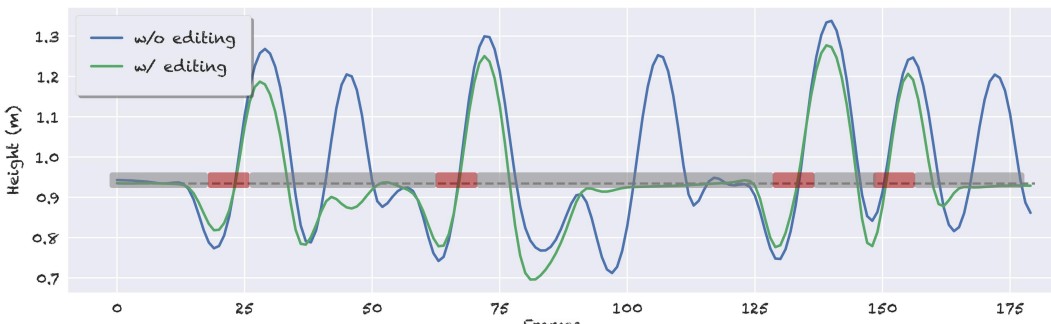

(a) The root height comparison. The red area denotes the timesteps to execute actions.

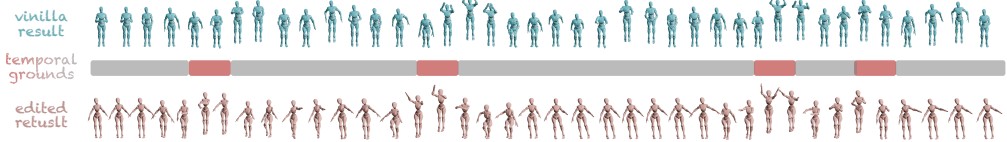

(b) The motion visualization. The vanilla generated result (blue) *vs*. edited result (red) w/ temporal grounds.

Figure 20: **Comparison between w/ *vs*. w/o grounded motion generation settings.** The root height and motion visualization of the textual prompt "a person jumps four times".

As depicted in Fig. 20, we provide a detailed comparison between the motion generation results with and without grounded settings. While the main text (Sec. 6) has already discussed the general differences between these settings, here in the appendix, we further extract and visualize the root height trajectory separately for a clearer and more detailed comparison. This approach helps in highlighting the effectiveness of our method in addressing motion hallucination issues and ensuring that the generated movements closely align with the given textual prompts.

In Fig. 20a, the root height comparison distinctly shows the difference between the edited and vanilla results. The red-shaded regions indicate the time steps where the specified actions ("jumps four times") should occur. Without grounded motion generation, the vanilla result tends to generate more than the required number of jumps, resulting in motion hallucination. However, with the incorporation of temporal grounding, our edited result accurately performs the action four times, aligning with the textual prompt. Fig. 20b further visualizes the motion sequences. It is evident that the temporal grounding guides the motion generation process, ensuring consistency with the input prompt. The edited result follows the correct sequence of actions, demonstrating the advantage of using grounded motion settings to avoid common hallucinations in generative models.

Overall, these detailed visualization results confirm the importance of incorporating temporal grounding into motion generation tasks, as it helps mitigate hallucinations in generative models, ensuring the generated motions are more faithfully aligned with the intended textual descriptions.

## B    USER INTERFACE FOR INTERACTIVE MOTION GENERATION AND EDITING

To have a better understanding of our task, we build a user interface with Gradio (Abid et al., 2019). We introduce the demo as follows.

In Fig. 21, we illustrate the steps involved in generating and visualizing motions using the interactive interface. Fig. 21a displays the initial step where the user provides input text such as "a man jumps" and adjusts motion parameters. Once the settings are finalized, the system begins processing the motion based on these inputs, as seen in the left panel. Fig. 21b showcases the generated motion based on the user's input. The interface provides a rendered output of the skeleton performing the described motion. This presentation allows users to easily correlate the input parameters with the resulting animation. The generated motion can further be edited by adjusting parameters such as the length of the motion, emphasizing or de-emphasizing certain actions, or replacing actions altogether, depending on user requirements. This process demonstrates how the interface facilitates a workflow from input to real-time motion visualization.

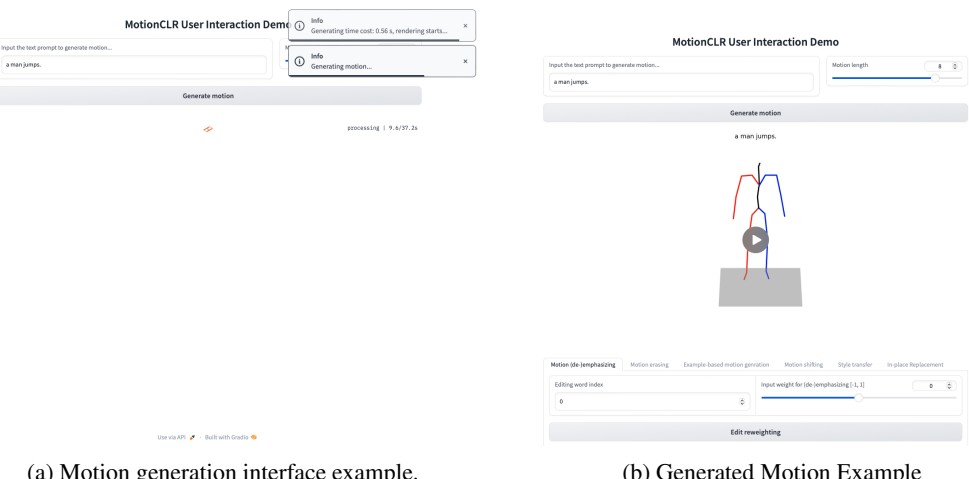

(a) Motion generation interface example.    (b) Generated Motion Example

Figure 21: Motion generation and its output examples.

The logical sequence of operations is as follows:

1. **Input the text:** Users start by entering text describing the motion (e.g., "a man jumps.") or set the frames of motions to generate (as shown in Fig. 21a).

2. **Generate the initial motion:** The system generates the corresponding skeleton motion sequence based on the input text (as shown in Fig. 21b).

3. **Motion editing:** We show some downstream tasks of MotionCLR here.

   - **Motion emphasizing/de-emphasizing:** Users can select a specific word from the text (e.g., "jumps") and adjust its emphasis using a weight slider (range [-1, 1]) (as seen in Fig. 22a). For example, setting the weight to 0.3 will either increase the jump motion's intensity.

   - **In-place replacement:** If users want to change the action, they can select the "replace" option. For example, replacing "jumps" with "walks" will regenerate the motion, showing a comparison between the original and new edited motions (as shown in Fig. 22b).

   - **Example-based motion generation:** Users can generate motion sequences based on predefined examples by setting parameters like chunk size and diffusion steps. After specifying the number of motions to generate, the system will create multiple variations of the input motion, providing diverse options for further refinement (as illustrated in Fig. 22d). The progress bars of the process are visualized in Fig. 22c.

**We leave a `interactive_demo.mp4` in the supplementary for demonstration.**

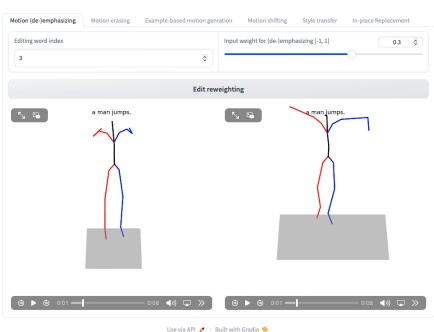

(a) Motion (de-)Emphasizing interface.

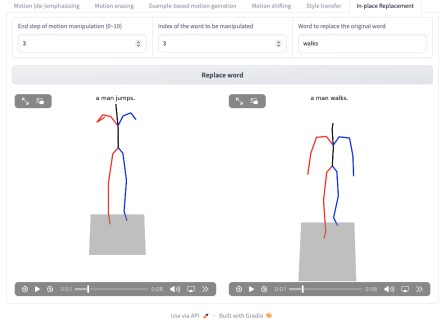

(b) In-place replacement example.

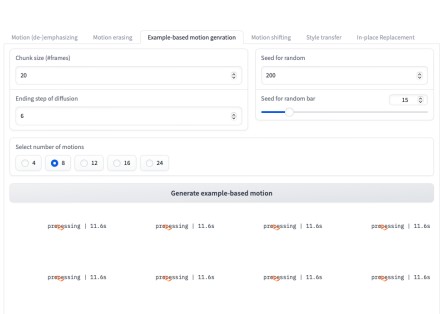

(c) Example-based motion generation progress.

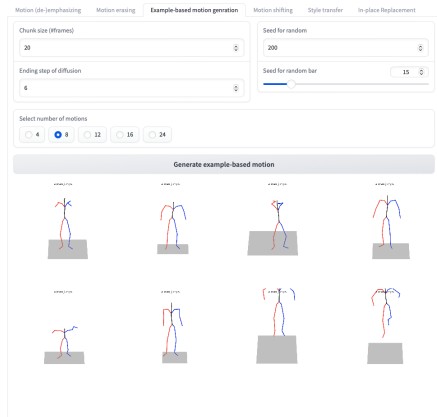

(d) Example-based motion generation results.

Figure 22: Different interfaces and supporting functions for interactive motion editing.

# C  DETAILED DIAGRAM OF ATTENTION MECHANISMS

## C.1  MATHEMATICAL VISUALIZATION OF SELF-ATTENTION MECHANISM

In the main text (Eq. (2)), we introduced the self-attention mechanism of MotionCLR, which utilizes different transformations of motion as inputs. The motion embeddings serve as both the query ($\mathbf{Q}$), key ($\mathbf{K}$), and value ($\mathbf{V}$), capturing the internal relationships within the sequence of motion frames.

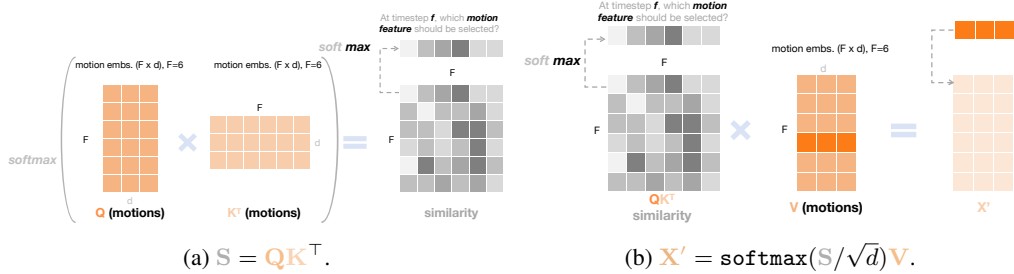

(a) $\mathbf{S} = \mathbf{Q}\mathbf{K}^\top$.        (b) $\mathbf{X'} = \texttt{softmax}(\mathbf{S}/\sqrt{d})\mathbf{V}$.

Figure 23: **Mathematical Visualization of Self-attention Mechanism.** This figure takes $F = 6$ as an example. (a) The similarity calculation with queries and keys (different frames). (b) The similarity matrix picks "value"s of the attention mechanism and updates motion features.

Fig. 23 provides a detailed mathematical visualization of this process:

(1) **Similarity Calculation**. In the first step, the similarity between the motion embeddings at different frames is computed using the dot product, represented by $\mathbf{S} = \mathbf{Q}\mathbf{K}^\top$. This measurement reflects the internal relationship/similarity between different motion frames within the sequence. Fig. 23a illustrates how the $\texttt{softmax}(\cdot)$ operation is applied to the similarity matrix to determine which motion feature should be selected at a given frame $f$.

(2) **Feature Updating**. Next, the similarity scores are used to weight the motion embeddings ($\mathbf{V}$) and generate updated features $\mathbf{X'}$, as shown by the equation $\mathbf{X'} = \texttt{softmax}(\mathbf{Q}\mathbf{K}^\top/\sqrt{d})\mathbf{V}$. Here, the similarity matrix applies its selection of values ($\mathbf{V}$) to update the motion features. This process allows the self-attention mechanism to dynamically adjust the representation of each motion frame based on its relevance to other frames in the sequence.

In summary, the self-attention mechanism aims to identify and emphasize the most relevant motion frames in the sequence, updating the features to enhance their representational capacity for downstream tasks. The most essential capability of cross-attention is to order the motion features.

## C.2  MATHEMATICAL VISUALIZATION OF CROSS-ATTENTION MECHANISM

In the main text (Eq. (3)), we introduced the cross-attention mechanism of MotionCLR, which utilizes the transformation of motion as a query ($\mathbf{Q}$) and the transformation of text as a key ($\mathbf{K}$) and value ($\mathbf{V}$) to explicitly model the correspondence between motion frames and words.

Fig. 24 provides a detailed mathematical visualization of this process:

(1) **Similarity Calculation**. In the first step, the similarity between the motion embeddings ($\mathbf{Q}$) with $F$ frames and the text embeddings ($\mathbf{K}$) with $N$ words is computed through the dot product, represented by $\mathbf{S} = \mathbf{Q}\mathbf{K}^\top$. This similarity measurement reflects the relationship between motion frames and words. Fig. 24a shows how the $\texttt{softmax}(\cdot)$ operation is applied to the similarity matrix to determine which word should be activated at a given frame $f$.

(2) **Feature Updating**. Next, the similarity scores are used to weight the text embeddings ($\mathbf{V}$) and generate updated features $\mathbf{X'}$, as shown by the equation $\mathbf{X'} = \texttt{softmax}(\mathbf{Q}\mathbf{K}^\top/\sqrt{d})\mathbf{V}$. Here, the similarity matrix applies its selection of values ($\mathbf{V}$) to update the features. This process establishes an explicit correspondence between the frames and specific words.

In summary, the similarity calculation process determines which frame(s) should be selected, and the feature updating process (multiplication with $\mathbf{V}$) is the execution of the frame(s) placement.

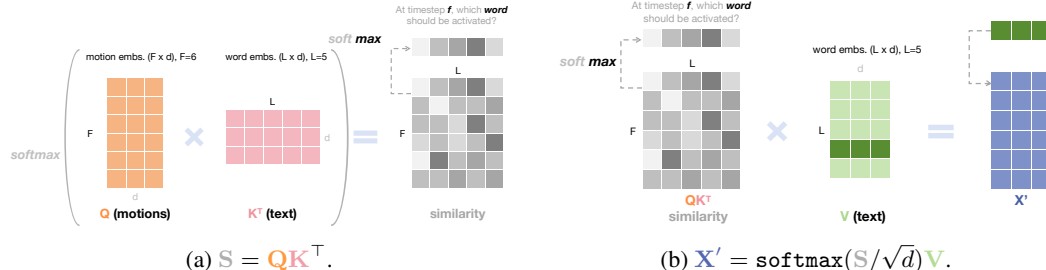

(a) $\mathbf{S} = \mathbf{Q}\mathbf{K}^{\top}$.

(b) $\mathbf{X}' = \texttt{softmax}(\mathbb{S}/\sqrt{d})\mathbf{V}$.

Figure 24: **Mathematical Visualization of Cross-attention Mechanism.** This figure takes $F = 6$ and $N = 5$ as an example. (a) The similarity calculation with queries and keys. (b) The similarity matrix picks "value"s of the attention mechanism and updates features.

### C.3 THE BASIC DIFFERENCE WITH PREVIOUS DIFFUSION-BASED MOTION GENERATION MODELS IN CROSS-MODAL MODELING

As discussed in the main text (see Sec. 1), despite the progresses in human motion generation (Zhang et al., 2024d; Cai et al., 2024; Zhang et al., 2024c; Guo et al., 2024c; Raab et al., 2024b; Kapon et al., 2024; Cohan et al., 2024; Fan et al., 2024; Xu et al., 2024; 2023a;b; Yao et al., 2022; Feng et al., 2023; Ao et al., 2023; Yao et al., 2024; Zhang et al., 2024f; Liu et al., 2010; Aberman et al., 2020a; Karunratanakul et al., 2024; Li et al., 2024; 2023a; Gong et al., 2023; Zhou & Wang, 2023; Zhong et al., 2023; Zhu et al., 2023; Athanasiou et al., 2023; Zhong et al., 2024; Guo et al., 2024b; Zhang et al., 2024c; Athanasiou et al., 2024; Zhao et al., 2023; Zhang et al., 2022; 2020; Diomataris et al., 2024; Pinyoanuntapong et al., 2024; Diller & Dai, 2024; Peng et al., 2023; Hou et al., 2023; Liu et al., 2023; Cong et al., 2024; Cui et al., 2024; Jiang et al., 2022; Kulkarni et al., 2024; Tessler et al., 2024; Liang et al., 2024; Ghosh et al., 2023; Wu et al., 2024), there still lacks a explicit modeling of word-level cross-modal correspondence in previous work. To clarify this, our method models a fine-grained word-level cross-modal correspondence.

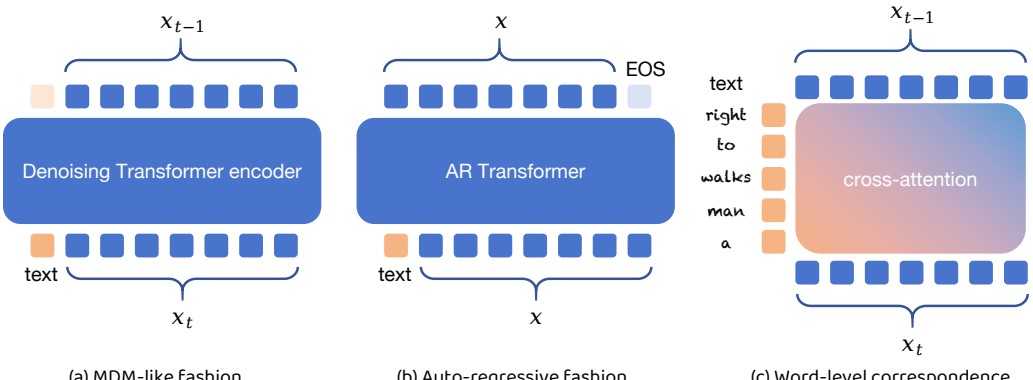

(a) MDM-like Fashion        (b) Auto-regressive fashion        (c) Word-level correspondence

Figure 25: **Comparison with previous diffusion-based motion generation models.** (a) MDM-like fashion: Tevet et al. (2022b) and its follow-up methods treat text embeddings as a whole and mix them with motion representations using a denoising Transformer. (b) Auto-regressive fashion: Zhang et al. (2023a) and its follow-up methods concatenate the text with the motion sequence and feed them into an auto-regressive transformer without explicit correspondence modeling. (c) Our proposed method establishes fine-grained word-level correspondence using cross-attention mechanisms.

As illustrated in Fig. 25, the major distinction between our proposed method and previous diffusion-based motion generation models lies in the explicit modeling of word-level cross-modal correspondence. In the MDM-like fashion Tevet et al. (2022b) (see Fig. 25a), previous methods usually utilize a denoising transformer encoder that treats the entire text as a single embedding, mixing it with the motion sequence. This approach lacks the ability to capture the nuanced relationship between individual words and corresponding motion elements, resulting in an over-compressed representation. Although we witness that Zhang et al. (2024b) also introduces cross-attention in the motion generation

process, it still faces two problems in restricting the fine-grained motion editing applications. First of all, the text embeddings are mixed with frame embeddings of diffusion, resulting in a loss of detailed semantic control. Our approach disentangles the diffusion timestep injection process in the convolution module to resolve this issue. Besides, the linear cross-attention in MotionDiffuse is different from the computation process of cross-attention, resulting in a lack of explanation of the word-level cross-modal correspondence. The auto-regressive (AR) fashion (Zhang et al., 2023a) (Fig. 25b) adopts a simple concatenation of text and motion, where an AR transformer processes them together. However, this fashion also fails to explicitly establish a fine-grained correspondence between text and motion, as the AR transformer merely regards the text and motion embeddings as one unified sequence.

Our approach (shown in Fig. 25c) introduces a cross-attention mechanism that explicitly captures the word-level correspondence between the input text and generated motion sequences. This allows our model to maintain a clear and interpretable mapping between specific words and corresponding motion patterns, significantly improving the quality and alignment of generated motions with the textual descriptions. By integrating such a word-level cross-modal modeling technique, our method not only achieves more accurate and realistic motion generation but also supports fine-grained word-level motion editing. This capability enables users to make precise adjustments to specific parts of the generated motion based on textual prompts, addressing the critical limitations present in previous diffusion-based motion generation models and allowing for more controllable and interpretable editing at the word level.

## D  MORE VISUALIZATION RESULTS OF EMPIRICAL EVIDENCE

In the main text, we introduced the foundational understanding of both cross-attention and self-attention mechanisms, emphasizing their ability to capture temporal relationships and dependencies across motion sequences. As a supplement, we provide a new, more detailed example here. As shown in Fig. 26, this visualization illustrates how different attention mechanisms respond to a complex sequence involving both walking and jumping actions. Specifically, we use green dashed boxes to highlight the "walk" phases and red dashed boxes to indicate the "jump" phases. This allows us to clearly distinguish the temporal patterns associated with each action. Besides, we observed that the word "jump" reaches its highest activation during the crouching phase, which likely correlates with this moment being both the start of the jumping action and the "power accumulation phase". This suggests that the attention mechanism accurately captures the preparatory stage of the movement, highlighting its capability to recognize the nuances of motion initiation within complex sequences. The cross-attention map effectively aligns key action words like "walk" and "jump" with their respective motion segments, while the self-attention map reveals repeated motion patterns and similarities between the walking and jumping cycles.

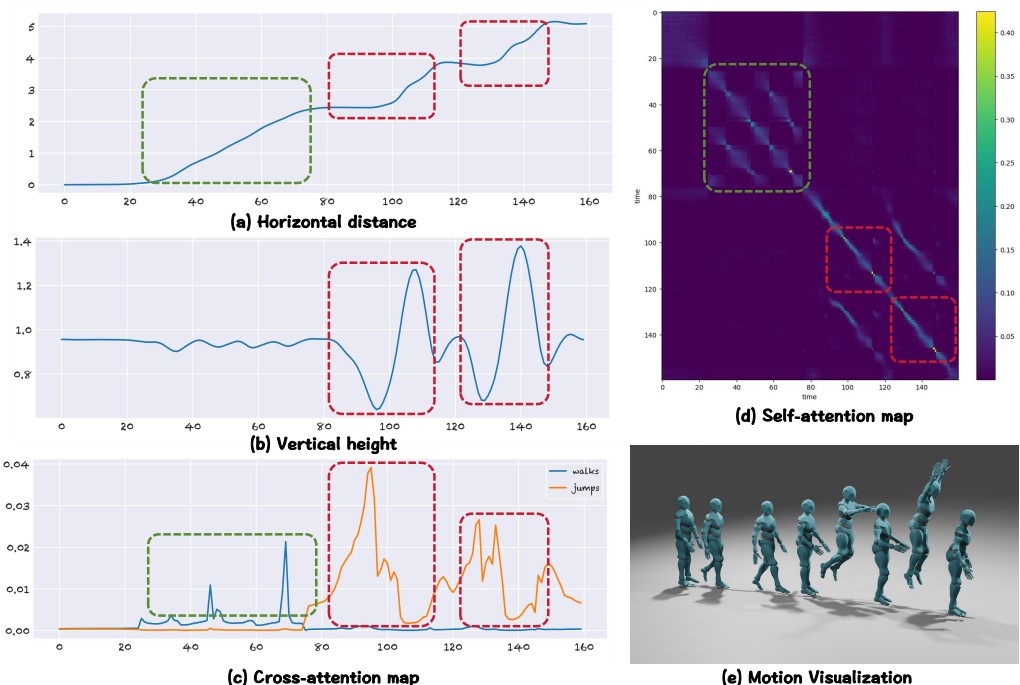

Figure 26: **Empirical study of attention patterns.** We use the example "a person walks stop and then jumps." (a) Horizontal distance traveled by the person over time, highlighting distinct walking and jumping phases. (b) The vertical height changes of the person, indicating variations during walking and jumping actions. (c) The **cross-attention** map between timesteps and the described actions. Notice that "walk" and "jump" receive a stronger attention signal corresponding to the walk and jump segments. (d) The **self-attention** map, which clearly identifies repeated walking and jumping cycles, shows similar patterns in the sub-actions. (e) Visualization of the motion sequences, demonstrating the walking and jumping actions.

Continuing with another case study, in Fig. 27, we examine how attention mechanisms respond to a sequence that primarily involves walking actions with varying intensity. In this instance, we observe that both the horizontal distance (Fig. 27a) and vertical height (Fig. 27b) reflect the man walks straight. The cross-attention map (Fig. 27c) reveals how the word "walks" related to walking maintains consistent activation, indicating that MotionCLR has a word-level understanding throughout the sequence. The self-attention map (Fig. 27d) further emphasizes repeated walking patterns, demonstrating that the mechanism effectively identifies the temporal consistency and structure of the walking phases. The motion visualization (Fig. 27e) reinforces this finding, showing a clear, uninterrupted walking motion.

More importantly, we can observe that the walking action consists of a total of five steps: three steps with the right foot and two with the left foot. The self-attention map (Fig. 27d) clearly reveals that steps taken by the same foot exhibit similar patterns, while movements between different feet show distinct differences. This observation indicates that the self-attention mechanism effectively captures the subtle variations between repetitive motions, further demonstrating its sensitivity to nuanced motion capture capability within the sequence.

Besides, different from the jumping, the highlights in the self-attention map of the walking are rectangular. The reason is that the local movements of walking are similar. In contrast, the jumping includes several sub-actions, resulting in the highlighted areas in the self-attention maps being elongated.

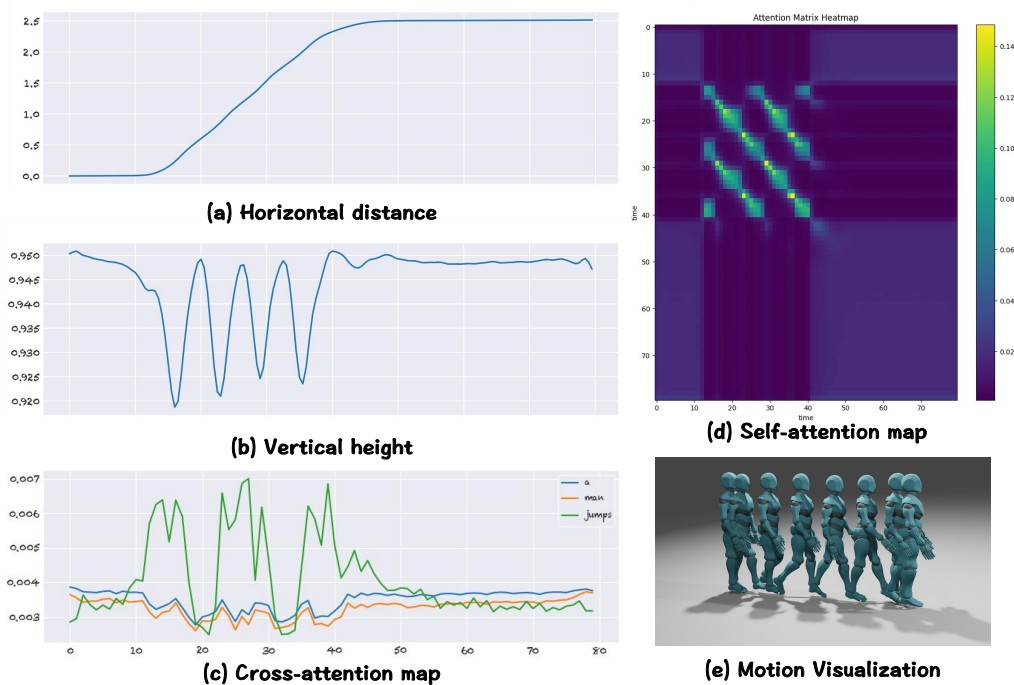

Figure 27: **Empirical study of attention patterns in a consistent walking sequence.** We use the example: "a man walks.". (a) The horizontal distance traveled over time reflects a steady walking motion. (b) Vertical height changes indicate minimal variation, characteristic of walking actions. (c) The **cross-attention** map shows that the "walks" word maintains consistent activation. (d) The **self-attention** map highlights the repeated walking cycles, capturing the temporal stability. (e) Visualization of the motion sequence.

# E    IMPLEMENTATION AND EVALUATION DETAILS

## E.1    IMPLEMENTATION DETAILS

The MotionCLR model is trained on the HumanML3D dataset with one NVIDIA A-100 GPU based on PyTorch (Paszke et al., 2019). The latent dimension of the motion embedding is 512. We take the CLIP-ViT-B model to encoder text as word-level embeddings. The training process utilizes a batch size of 64, with a learning rate initialized at $2e-4$ and decaying at a rate of $0.9$ every $5,000$ steps. Additionally, a weight decay of $1e-2$ is employed to regularize the model parameters. For the diffusion process, the model is trained over $1,000$ diffusion steps. We incorporate a probability of $0.1$ for condition masking to facilitate classifier-free guidance learning. During training, dropout is set at $0.1$ to prevent overfitting, and all networks in the architecture follow an 8-layer Transformer design.

In the inference stage, all steps of the denoising sampling are set as $10$ consistently. For the motion erasing application, we set the erasing weight as 0.1 as default. MotionCLR supports both DDIM-sampling (Song et al., 2021) and DPM-soler-sampling (Lu et al., 2022) methods, with $1,000$ as full diffusion steps. For the in-placement motion replacement and the motion style transfer application, as the motion semantics mainly depend on the initial denoising steps, we set the manipulating steps until 5 as default. For motion (de-)emphasizing, we support both multiplications (larger than 1 for emphasizing, lower than 1 for de-emphasizing) and addition (larger than 0 for emphasizing, lower than 0 for de-emphasizing) to adjust the cross-attention weights. For the example-based motion generation, the minimum manipulating time of a motion zone is 1s (*a.k.a.* chunk size=20 for the 20 FPS setting). At each step, all attention maps at all layers will be manipulated at each denoising timestep. Users can adjust the parameters freely to achieve interactive motion generation and editing (more details of user interface in Appendix B).

## E.2    COMPARED BASELINES

Here, we introduce details of baselines in Tab. 1 for our comparison.

**TM2T** (Guo et al., 2022b) explores the reciprocal generation of 3D human motions and texts. It uses motion tokens for compact representation, enabling flexible generation for both text2motion and motion2text tasks.

**T2M** (Guo et al., 2022a) generates diverse 3D human motions from text using a two-stage approach involving text2length sampling and text2motion generation. It employs a motion snippet code to capture semantic contexts for more faithful motion generation.

**MDM** (Tevet et al., 2022b) uses a diffusion-based approach with a transformer-based design for generating human motions. It excels at handling various generation tasks, achieving satisfying results in text-to-motion tasks.

**MLD** (Chen et al., 2023b) uses a diffusion process on motion latent space for conditional human motion generation. By employing a Variational AutoEncoder (VAE), it efficiently generates vivid motion sequences while reducing computational overhead.

**MotionDiffuse** (Zhang et al., 2024b) is a diffusion model-based text-driven framework for motion generation. It provides diverse and fine-grained human motions, supporting probabilistic mapping and multi-level manipulation based on text prompts.

**T2M-GPT** (Zhang et al., 2023a) combines a VQ-VAE and GPT to generate human motions from the text. With its simple yet effective design, it achieves competitive performance and outperforms some diffusion-based methods on specific metrics.

**ReMoDiffuse** (Zhang et al., 2023b) integrates retrieval mechanisms into a diffusion model for motion generation, enhancing diversity and consistency. It uses a Semantic-Modulated Transformer to incorporate retrieval knowledge, improving text-motion alignment.

**MoMask** (Guo et al., 2024a) introduces a masked modeling framework for 3D human motion generation using hierarchical quantization. It outperforms other methods in generating motions and is applicable to related tasks without further fine-tuning.

### E.3 EVALUATION DETAILS

**Motion (de-)emphasizing.** To evaluate the effectiveness of motion (de-)emphasizing application, we construct 100 prompts to verify the algorithm. All of these prompts are constructed by researchers manually. We take some samples from our evaluation set as follows.

```
... ...
3 the figure leaps high
4 a man is waving hands
... ...
```

Each line in the examples represents the index of the edited word in the sentence, followed by the corresponding prompt. These indices indicate the key verbs or actions that are subject to the (de-)emphasizing during the evaluation process. The prompts were carefully selected to cover a diverse range of actions, ensuring that our method is tested on different types of motion descriptions. For instance, in the prompt "3 the figure leaps high", the number 3 indicates that the word "leaps" is the third word in the sentence and is the target action for (de-)emphasizing. This format ensures a systematic evaluation of how the model responds to adjusting attention weights on specific actions across different prompts.

**Example-based motion generation.** To further evaluate our example-based motion generation algorithm, we randomly constructed 7 test prompts. We used t-SNE (Pedregosa et al., 2011) visualization to analyze how closely the generated motions resemble the provided examples in terms of motion textures. For each case, the generated motion was assessed based on two criteria: (1) similarity to the example, and (2) diversity across different generated results from the same example.

**Action counting.** To thoroughly evaluate the effectiveness of our action counting method, we constructed a test set containing 70 prompts. These prompts were manually designed by researchers to ensure diversity. Each prompt corresponds to a motion sequence generated by our model, and the ground truth action counts were labeled by researchers based on the observable actions within the generated motions.

## F DETAILS OF MOTION EDITING

In this section, we will provide more technical details about the motion editing algorithms.

### F.1 PSEUDO CODES OF MOTION EDITING

**Motion (de-)emphasizing.** Motion (de-)emphasizing mainly manipulate the cross-attention weights of the attention map. Key codes are shown in the `L16-18` of Code 1.

```python
def forward(self, x, cond, reweighting_attn, idxs):
    B, T, D = x.shape
    N = cond.shape[1]
    H = self.num_head

    # B, T, 1, D
    query = self.query(self.norm(x)).unsqueeze(2).view(B, T, H, -1)
    # B, 1, N, D
    key = self.key(self.text_norm(cond)).unsqueeze(1).view(B, N, H, -1)

    # B, T, N, H
    attention = torch.einsum('bnhd,bmhd->bnmh', query, key) / math.sqrt(D
        // H)
    weight = self.dropout(F.softmax(attention, dim=2))

    # reweighting attention for motion (de-)emphasizing
    if reweighting_attn > 1e-5 or reweighting_attn < -1e-5:
        for i in range(len(idxs)):
            weight[i, :, 1 + idxs[i]] = weight[i, :, 1 + idxs[i]] +
                reweighting_attn

    value = self.value(self.text_norm(cond)).view(B, N, H, -1)
    y = torch.einsum('bnmh,bmhd->bnhd', weight, value).reshape(B, T, D)
    return y
```

Code 1: Pseudo codes for motion (de-)emphasizing.

**In-place motion replacement.** The generation of two motions (`B=2`) are reference and edited motions. As the cross-attention map determines when to execute the action. Therefore, replacing the cross-attention map directly is a straightforward way, which is shown in `L16-17` of Code 2.

```python
def forward(self, x, cond, manipulation_steps_end):
    B, T, D = x.shape
    N = cond.shape[1]
    H = self.num_head

    # B, T, 1, D
    query = self.query(self.norm(x)).unsqueeze(2).view(B, T, H, -1)
    # B, 1, N, D
    key = self.key(self.text_norm(cond)).unsqueeze(1).view(B, N, H, -1)

    # B, T, N, H
    attention = torch.einsum('bnhd,bmhd->bnmh', query, key) / math.sqrt(D
        // H)
    weight = self.dropout(F.softmax(attention, dim=2))

    # replacing the attention map directly
    if self.step <= manipulation_steps_end:
        weight[1, :, :, :] = weight[0, :, :, :]

    value = self.value(self.text_norm(cond)).view(B, N, H, -1)
    y = torch.einsum('bnmh,bmhd->bnhd', weight, value).reshape(B, T, D)
    return y
```

Code 2: Pseudo codes for in-place motion replacement.

**Motion sequence shifting.** Motion sequence shifting aims to correct the atomic motion in the temporal order you want. We only need to shift the temporal order of $\mathbf{Q}$s, $\mathbf{K}$s, and $\mathbf{V}$s in the self-attention to obtain the shifted result. Key codes are shown in the `L13-24` and `L32-36` of Code 3.

```python
def forward(self, x, cond, time_shift_steps_end, time_shift_ratio):
    B, T, D = x.shape
    H = self.num_head

    # B, T, 1, D
    query = self.query(self.norm(x)).unsqueeze(2)
    # B, 1, T, D
    key = self.key(self.norm(x)).unsqueeze(1)
    query = query.view(B, T, H, -1)
    key = key.view(B, N, H, -1)

    # shifting queries and keys
    if self.step <= time_shift_steps_end:
        part1 = int(key.shape[1] * time_shift_ratio)
        part2 = int(key.shape[1] * (1 - time_shift_ratio))
        q_front_part = query[0, :part1, :, :]
        q_back_part = query[0, -part2:, :, :]
        new_q = torch.cat((q_back_part, q_front_part), dim=0)
        query[1] = new_q

        k_front_part = key[0, :part1, :, :]
        k_back_part = key[0, -part2:, :, :]
        new_k = torch.cat((k_back_part, k_front_part), dim=0)
        key[1] = new_k

    # B, T, N, H
    attention = torch.einsum('bnhd,bmhd->bnmh', query, key) / math.sqrt(D
        // H)
    weight = self.dropout(F.softmax(attention, dim=2))
    value = self.value(self.text_norm(cond)).view(B, T, H, -1)

    # shifting values
    if self.step <= time_shift_steps_end:
        v_front_part = value[0, :part1, :, :]
        v_back_part = value[0, -part2:, :, :]
        new_v = torch.cat((v_back_part, v_front_part), dim=0)
        value[1] = new_v
    y = torch.einsum('bnmh,bmhd->bnhd', weight, value).reshape(B, T, D)
    return y
```

Code 3: Pseudo codes for motion sequence shifting.

**Example-based motion generation.** To generate diverse motions driven by the same example, we only need to shuffle the order of queries in self-attention, which is shown in L13-23 of Code 4.

```python
def forward(self, x, cond, steps_end, _seed, chunk_size, seed_bar):
    B, T, D = x.shape
    H = self.num_head

    # B, T, 1, D
    query = self.query(self.norm(x)).unsqueeze(2)
    # B, 1, T, D
    key = self.key(self.norm(x)).unsqueeze(1)
    query = query.view(B, T, H, -1)
    key = key.view(B, N, H, -1)

    # shuffling queries
    if self.step == steps_end:
        for id_ in range(query.shape[0]-1):
            with torch.random.fork_rng():
                torch.manual_seed(_seed)
                tensor = query[0]
                chunks = torch.split(tensor, chunk_size, dim=0)
                shuffled_index = torch.randperm(len(chunks))
                shuffled_chunks = [chunks[i] for i in shuffled_index]
                shuffled_tensor = torch.cat(shuffled_chunks, dim=0)
                query[1+id_] = shuffled_tensor
                _seed += seed_bar

    # B, T, T, H
    attention = torch.einsum('bnhd,bmhd->bnmh', query, key) / math.sqrt(D
        // H)
    weight = self.dropout(F.softmax(attention, dim=2))
    value = self.value(self.text_norm(cond)).view(B, N, H, -1)
    y = torch.einsum('bnmh,bmhd->bnhd', weight, value).reshape(B, T, D)
    return y
```

Code 4: Pseudo codes for example-based motion generation.

**Motion style transfer.** In the generation of two motions (B=2), we only need to replace the query of the second motion with the first one, which is shown in L13-14 of Code 5.

```python
def forward(self, x, cond, steps_end):
    B, T, D = x.shape
    H = self.num_head

    # B, T, 1, D
    query = self.query(self.norm(x)).unsqueeze(2)
    # B, 1, T, D
    key = self.key(self.norm(x)).unsqueeze(1)
    query = query.view(B, T, H, -1)
    key = key.view(B, N, H, -1)

    # style transfer
    if self.step <= self.steps_end:
        query[1] = query[0]

    # B, T, T, H
    attention = torch.einsum('bnhd,bmhd->bnmh', query, key) / math.sqrt(D
        // H)
    weight = self.dropout(F.softmax(attention, dim=2))
    value = self.value(self.text_norm(cond)).view(B, N, H, -1)
    y = torch.einsum('bnmh,bmhd->bnhd', weight, value).reshape(B, T, D)
    return y
```

Code 5: Pseudo codes for motion style transfer.

## F.2 SUPPLEMENTARY FOR MOTION STYLE TRANSFER

As discussed in the main text, motion style transfer is accomplished by replacing the query ($\mathbf{Q}$) from the content sequence ($\mathbf{M_2}$) with that from the style sequence ($\mathbf{M_1}$). This replacement ensures that while the content features from $\mathbf{M_2}$ are preserved, the style features from $\mathbf{M_1}$ are adopted, resulting in a synthesized motion sequence that captures the style of $\mathbf{M_1}$ with the content of $\mathbf{M_2}$.

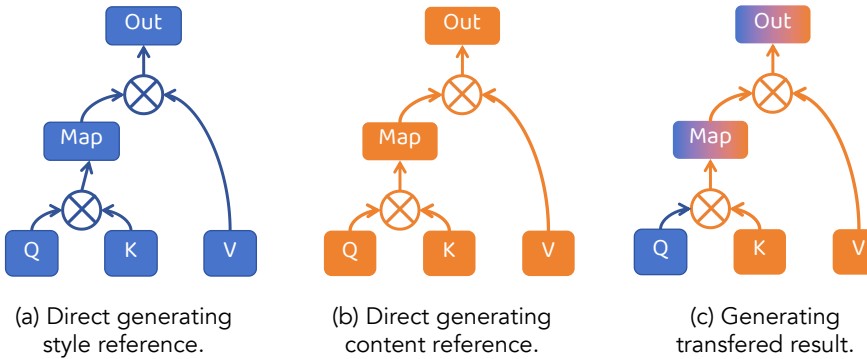

(a) Direct generating
style reference.

(b) Direct generating
content reference.

(c) Generating
transfered result.

Figure 28: The illustration of motion style transfer process. (a) Direct generating style reference: The style information is generated directly using the query ($\mathbf{Q}$), key ($\mathbf{K}$), and value ($\mathbf{V}$) from the style reference motion sequence (blue). (b) Direct generating content reference: The content information is generated directly from the content reference motion sequence (orange). (c) Generating transferred result: The final transferred motion sequence combines the style from the style reference sequence with the content from the content reference sequence, using $\mathbf{Q}$ from the style reference (blue) and $\mathbf{K}$, $\mathbf{V}$ from the content reference (orange).

Fig. 28 provides a visual explanation of this process. The self-attention mechanism plays a crucial role, where the attention map determines the correspondence between the style and content features. The pseudo code snippet provided in Code 5 exemplifies this process. By setting "`query[1] = query[0]`" in the code, the query for the second motion ($\mathbf{M_2}$) is replaced by that of the first motion ($\mathbf{M_1}$), which effectively transfers the motion style from $\mathbf{M_2}$ to $\mathbf{M_1}$. In summary, this motion style transfer method allows one motion sequence to adopt the style characteristics of another while maintaining its own content.

## G    DETAILS OF ACTION COUNTING IN A MOTION

The detailed process of action counting is described in Code 6. The attention map is first smoothed using a Gaussian filter to eliminate noise, ensuring that minor fluctuations do not affect peak detection. We then downsample the smoothed matrix to reduce computational complexity and normalize it within a 0-1 range for consistent peak detection across different motions.

The pseudo code provided demonstrates the complete process, including peak detection using height and distance thresholds. The experimental results indicate that this approach is more reliable and less sensitive to noise compared to using the root trajectory, thus confirming the effectiveness of our method in accurately counting actions within a generated motion sequence.

```python
"""
Input: matrix (the attention map array with shape (T, T))
Output: float (counting number)
"""

# Apply Gaussian smoothing via gaussian_filter in scipy.ndimage
smoothed_matrix = gaussian_filter(matrix, sigma=0.8)

# Attention map down-sampling
downsample_factor = 4
smoothed_matrix = downsample_matrix(smoothed_matrix, downsample_factor)

# Normalize the matrix to 0-1 range
normalized_matrix = normalize_matrix(smoothed_matrix)

# Detect peaks with specified height and distance thresholds
height_threshold = normalized_matrix.mean() * 3 # you can adjust this
distance_threshold = 1  # you can adjust this
peaks_positions_per_row = detect_peaks_in_matrix(normalized_matrix,
    height=height_threshold, distance=distance_threshold)

# Display the peaks positions per row
total_peak = sum([len(i) if len(i) > 0 else 0 for i in
    peaks_positions_per_row])
sum_ = sum([1 if len(i) > 0 else 0 for i in peaks_positions_per_row])

return total_peak / sum_
```

Code 6: Pseudo codes for action counting.

**Evaluation on alignment between attention maps and actions.** Given that our work represents an early exploration into the area of motion editing through manipulation of cross-/self-attention, a comprehensive evaluation protocol for this task is still hard in the research community. Despite this limitation, we have made efforts to develop a preliminary quantitative evaluation to bridge this gap.

To better quantify the alignment between attention weights and motion, we employ the Intersection over Union (IoU) metric. The IoU metric is used to measure the overlap between regions of high attention and regions of significant motion intensity, defined as follows.

- We consider attention values above 65% of the maximum value as indicating active regions associated with specific actions.
- Similarly, we define active regions in root velocity based on the intensity of motion.
- The IoU is calculated between the attention-derived active regions and the corresponding motion intensity regions, providing a measure of temporal correspondence.

Table 8 presents the IoU results under different temporal shifts, demonstrating a strong alignment between the attention weights and the motion execution areas.

The IoU metric serves as a complementary evaluation to the action counting metric discussed in Sec. 5.4. The high IoU values indicate a good temporal correspondence between attention weights

| adjusting weight | -0.1 | 0 | +0.1 |
|:---:|:---:|:---:|:---:|
| IoU (%) | 74.3 | 75.5 | 76.2 |

Table 8: IoU values for alignment between attention maps and actions under different temporal areas.

and the execution of actions, thereby enhancing the quantitative assessment of our proposed motion manipulation approach.

We believe that the development of more advanced metrics in the future would further benefit the evaluation of motion editing and attention-based motion manipulation. This initial exploration lays the groundwork for more comprehensive assessment methods in subsequent research.

