# OpenReview forum: "MotionCLR: Motion Generation and Training-free Editing via Understanding Attention Mechanisms"
_ICLR.cc/2025/Conference — Submitted to ICLR 2025_

### Official Review · Reviewer_hAdU · 2024-11-01

**Soundness:** 4
**Presentation:** 4
**Contribution:** 4
**Rating:** 10
**Confidence:** 4

**Summary:**

This paper presents MotionCLR, an attention-based motion diffusion model that enables fine-grained motion generation and editing. The key innovation lies in its explicit modeling of word-level text-motion correspondence through a dedicated cross-attention mechanism while using self-attention to capture temporal relationships between motion frames.

The model achieves various motion editing capabilities without requiring paired editing data, including motion emphasis/de-emphasis, in-place motion replacement, example-based generation, and style transfer. The authors provide both theoretical analysis and empirical evidence to demonstrate how the attention mechanisms enable these editing capabilities.

Extensive experiments show that MotionCLR achieves competitive performance on standard motion generation benchmarks while offering editing flexibility. The paper also explores the model's potential for action counting and addressing hallucination issues in motion generation.

**Strengths:**

- Novel technical contribution in disentangling text-motion correspondence and motion-motion interactions through separate attention mechanisms
- Thorough analysis and validation of how attention mechanisms enable motion control
- Impressive range of editing capabilities achieved without paired editing data
- Strong quantitative results compared to state-of-the-art methods

**Weaknesses:**

- The cross-attention control method used for motion editing comes from an existing text-to-image editing paper, Prompt-to-Prompt (Hertz et al., 2023). Although this reference is cited, the connection between this submission and Prompt-to-Prompt should be clarified to avoid misunderstanding.
- The authors did not compare the performance of motion editing against other methods. It would be impressive to show that this method performs similarly to those that require paired data and training.

**Questions:**

- When conducting cross-attention control, do you manipulate the attention weights on all cross-attention layers in all CLR modules or only in some specific layers?
- In the emphasize and de-emphasize cases, why are the weights added and not multiplied by the adjustment?
- We see a significant bump in FID when the editing weight moves beyond +/- 0.5 in Table 2. What might be the reasons behind this phenomenon?

---

> ### Author Response · Authors · 2024-11-17
>
> We authors team sincerely acknowledge your insightful suggestions and positive feedback on this work. We would include reviewer `hAdU` in our acknowledgment if accepted due to their constructive suggestions. We would like to resolve your concerns as follows.
>
> ---
>
> **Q1**: Discussing with Hertz et al. (2023).
>
> **A1**: Thanks for pointing out this! We acknowledge the technical details of motion (de-)emphasizing and replacement is similar to the method in P2P. However, P2P does not show how to introduce self-attention into the editing process, which is what our method can do. Besides, introducing related techniques into the animation community and fine-grained explainability on attentions has also not been observed. **We have revised accordingly in the PDF.**
>
> **Q2**: Other motion editing baselines.
>
> **A2**: We are happy to discuss related baselines. We noticed that there are limited methods that can support interactive motion editing. We find the most related work is MotionFix (SIGGRAPH Asia 2024, to appear in Dec. 2024). MotionFix was uploaded to Arxiv in Aug. 2024 and the ICLR submission deadline is Oct.-01 2024. We treat this as a concurrent work. We also cite it in related work.
>
> **MotionFix needs human annotation** from Amazon Mechanical Turk (AMT). As the *interaction fashion* is different from ours, we test some cases of MotionFix and provide the feedback to reviewer. MotionFix cannot achieve style transfer, replacement, and example-based motion generation, due to limited/no annotation about these tasks. Although their method can achieve some editing results like “jump higher”, MoitonFix cannot control the jumping times of the result. This is mainly due to its limited data annotations.
>
> In contrast, we would like to **highlight** that our MotionCLR is promising to build a motion editing dataset in a very **scalable** way, which is what we are now working on.
>
> **Q3**: Attention manipulation layers.
>
> **A3**: This is an **insightful** question! We manipulate all attention layers and find it is the best setting. We infer the reason is that all attention matters. We provide the ablation in **Appendix A.3** (Tab. 6) in our original submission.
>
> **Q4**: Weight adjustment ways.
>
> **A4**: Thanks for the detailed question! MotionCLR supports both adding and multiplication editing modes, which **were specified in Appendix E.1** of the original submission. For convenient presentation, we provide only adding mode for presentation.
>
> **Q5**: FID of manipulating results.
>
> **A5**: The observation is right. We specified this question in Sec. 5.2 of the original paper: “When adjusting weights are too large, the attention maps are corrupted, resulting in artifacts.”. We hope this will resolve your question.
>
> ---
>
> We would like to acknowledge reviewer `hAdU` again for their such positive recommendation, which encourages us a lot!

---

> > ### Author Response · Authors · 2024-11-20
> >
> > Dear Reviewer `hAdU`:
> >
> > Thanks again for your efforts in reviewing! We are looking forward to your reply. Would you mind checking our response and confirming if there are unclear explanations?
> >
> > We are highly encouraged if your concerns have been addressed. If you need any more clarification, we can provide it as soon as possible before the discussion deadline.
> >
> > Best,
> >
> > MotionCLR author(s) team

---

> > > ### Comment · Reviewer_hAdU · 2024-11-22
> > >
> > > Thank you for the detailed response. Despite other reviewers comments, I remain positive on my position of this paper. The attention based modeling and editing of human motion generation is a significant advance to the community.

---

> > > > ### Author Response · Authors · 2024-11-22
> > > >
> > > > Dear reviewer `hAdU`,
> > > >
> > > > Thanks again for your efforts in reviewing this work and providing such a positive review on it. We really appreciate it and we believe this work will motivate follow-up research to explore interactive motion editing.
> > > >
> > > > Best,
> > > >
> > > > MotionCLR author(s) team

---

### Official Review · Reviewer_b8bV · 2024-11-02

**Soundness:** 2
**Presentation:** 4
**Contribution:** 3
**Rating:** 3
**Confidence:** 4

**Summary:**

The authors proposed a method for generating and editing motion using a text-aligned motion generation process. By exploring attention mechanisms, they discovered that manipulating attention weights enables motion editing. Their approach can be applied to downstream tasks such as motion emphasizing and de-emphasizing, motion erasing, and motion sequence shifting.

**Strengths:**

The paper demonstrated that the cross-attention mechanism plays a crucial role in establishing text-motion correspondence at the word level.
This insight enables a better understanding and manipulation of text-to-motion models with finer granularity.
The proposed method significantly improves the explainability of text-to-motion generation methods.

**Weaknesses:**

While I believe the paper will play an important role in the development of text-to-motion generation task, I have serious concerns regarding its evaluation method.

In my view, the experimental evaluation relies too heavily on qualitative and empirical results, including attention map visualizations and velocity/trajectory visualizations. The action counting error metric currently used only indicates the number of actions and fails to capture the **temporal correspondence** between the attention map and human motion. Given that the paper's primary contribution is editing motion in by manipulating attention weights, it is crucial that the evaluation demonstrates temporal correspondence between these attention weights and the motion itself. The authors could have proposed an evaluation metric measuring the correspondence or similarity between the attention weights—whether self-attention or cross-attention, depending on the downstream task—and motion intensity over time, which would better elucidate the manipulation. The TMR metric, which calculates the similarity between motion and text, somewhat makes sense for evaluating motion Motion (de-)emphasizing and motion erasing but I don't think it's enough to explain the temporal correspondence. Metrics like TMR and FID are insufficient to fully assess the manipulation either. I believe the quantitative evaluation of motion manipulation in the paper could be improved.

**Questions:**

In paragraph 5.2, it is stated that TMR-sim values range between 0 and 1, but in Table 2, the results are numbers like 55. This discrepancy requires additional explanation. Considering the inconsistencies among Tables 2, 4, 5, and 6, it appears that the results in Tables 2 and 4 might have been multiplied by 100 to highlight the differences. If so, this scaling should be clearly indicated to ensure consistency and avoid confusion.

---

> ### Author Response · Authors · 2024-11-17
>
> We appreciate the contribution to handling this submission and the constructive feedback. As this is an early attempt at motion editing, we do agree that this work will play an important role in the community. We would like to resolve your concerns as follows.
>
> ---
>
> **Q1**: About the evaluation protocol.
>
> **A1**: Your suggestion on the evaluation is quite insightful! We would like to resolve this concern from two aspects.
>
> - As this work is **an early exploration** of motion editing and the early work exploring the explainability of cross-/self-attention in motion generation, the community still lacks a thorough evaluation protocol to evaluate the task.
> - Although we treat **this might be acceptable and tolerable** for early exploration in the community, we try to **additionally evaluate** the alignment between the attention map with the root velocity **via the IoU metric** (Intersection over Union). For example, if the attention value of the attention is larger than 65% of the maximum attention values, we treat the word of action as activated. Similarly, we also applied this activation assessment on the velocity and found the execution area of actions. We calculate the IoU value of two areas as follows.
>
>
>     | adjusting weight | -0.1 | 0 | +0.1 |
>     | --- | --- | --- | --- |
>     | IoU (%) | 74.3% | 75.5% | 76.2% |
>
> We treat the results as complementary to the action counting result in Sec. 5.4. The IoU values show a good alignment between attention values and the action execution.
>
> **We have revised accordingly in the PDF (Appendix G)**. If reviewers have more suggestions on evaluation metrics, we are happy to include them, which will be useful to our community. **We will also include reviewer b8bV in our acknowledgment if accepted, due to this constructive suggestion.**
>
> **Q2**: About scales of TMR-sim..
>
> **A2**: Thanks for so detailed checking. You are right, we re-clarified them in the PDF.
>
> ---
>
> Dear reviewer `b8bV`, if your concerns are resolved, could you please consider **reevaluating this work and improve your rating**? If not or you have further guidance, we are happy to resolve them.

---

> > ### Author Response · Authors · 2024-11-20
> >
> > Dear Reviewer `b8bV`:
> >
> > Thanks again for your efforts in reviewing. We are looking forward to your reply. Would you mind checking our response and confirming if there are unclear explanations?
> >
> > We are highly encouraged if your concerns have been addressed. If your concerns are resolved, could you please consider reevaluating this work and improving your rating? If you need any more clarification, we can provide it as soon as possible before the discussion deadline.
> >
> > Best,
> >
> > MotionCLR author(s) team

---

> > ### Comment · Reviewer_b8bV · 2024-11-22
> >
> > I appreciate the authors' response to my comments and highly value their effort to convey an important message to the community through extensive experiments.
> >
> > While I understand and respect the authors' perspective that TMR-sim can be considered an acceptable metric, it does not seem to sufficiently support their arguments. Additionally, framing this work as an early exploration does not, in my view, justify the reliance on an indirect evaluation metric to demonstrate the paper's effectiveness.
> >
> > IoU appears more reasonable compared to the other metrics discussed; however, the authors’ claims would benefit from being supported by a more appropriate evaluation metric. This remains important even if the work is in its early stages and the proposed metric is not the primary contribution of the paper.
> >
> > That said, as noted in my original comments, the paper conveys an important message and has the potential to make a meaningful impact on the field. While I am inclined to raise my score, I still have minor concerns about the paper. Therefore, I will finalize my score after the discussion between the authors and reviewer DjgQ concludes.

---

> > > ### Author Response · Authors · 2024-11-24
> > > **Additional results to b8bV**
> > >
> > > Dear reviewer `b8bV`,
> > >
> > > Sorry to bother you again. As `DjgQ` has not joined our discussion so far, we would like to provide latest experiment results here to support our claim, **which we think is more convincing!**
> > >
> > > We noticed that **TMR supports a function of “moment retrieval”**, which supports calculating the text-action similarity between *text* and *a motion subinterval* (*e.g.* you can treat it as a window size of 20 frames). For example, for a motion generated by “a man walks forward”, we can calculate the similarity between the word “walks” and all 20-frame subintervals along the sequence. As a result, we can obtain the word-action similarity along the sequence.
> > >
> > > Based on this, we provide IoU comparisons among cross-attention, moment-retrieval-value, and root velocity. The setting is similar to the previous response. The results are as follows.
> > >
> > > |Exp. ID  | adjusting weight | -0.1 | 0 | +0.1 |
> > > | --- | --- | --- | --- | --- |
> > > | E1 (IoU, %)  | cross-attention & root velocity | 74.3% | 75.5% | 76.2% |
> > > | E2  (IoU, %) | cross-attention & moment-retrieval-value | 72.5% | 73.2% | 73.1% |
> > > | E3 (IoU, %)  | root velocity & moment-retrieval-value | 76.5% | 77.2% | 77.4% |
> > >
> > > We notice that the relation between cross-attention and the other two metrics (E1, E2) is very close to the similarity between kinematics and semantics (E3). This verifies that cross-attention actually models the fine-grained correspondence in both kinematics and semantic levels.
> > >
> > > If you have any confusion about the moment retrieval in [TMR](https://arxiv.org/pdf/2305.00976), a quick look at Fig. 4 in [TMR](https://arxiv.org/pdf/2305.00976) will be helpful to your understanding.
> > >
> > > We thank your suggestions on this and benefit a lot from this. We always value the suggestions from reviewers significantly and are looking for further updates.
> > >
> > > Best,
> > >
> > > MotionCLR author(s) team

---

> > > > ### Comment · Reviewer_b8bV · 2024-11-24
> > > >
> > > > Thank you for the additional information; I will take it into account when finalizing my evaluation.
> > > >
> > > > However, I am unable to find a clear explanation of the cross-attention manipulation. Could you clarify whether the authors applied the attention weight adjustment to the attention map or the similarity matrix? In Figure 3-(c), the attention map does not appear to exhibit the sum-to-one property along the text dimension. Additionally, unlike the use of the term “self-attention map,” the authors seem to have intentionally avoided using the term “cross-attention map.” Does this imply that the manipulation was instead applied to the similarity matrix?

---

> ### Author Response · Authors · 2024-11-22
> **Thanks for your discussion!**
>
> Dear reviewer `b8bV`,
>
> Thanks for your in-depth and insightful discussion! We are happy to know your high evaluations of our efforts!
>
> We do understand your views on the quantitative metrics. For more clarification, we would like to share some evaluations in attention-based image editing areas, like [MasaCtrl](https://arxiv.org/pdf/2304.08465) (ICCV-23), [PnP](https://arxiv.org/pdf/2211.12572) (CVPR-23), and [MoA](https://arxiv.org/pdf/2404.11565) (SIGGRAPH-24). **Most of these methods rely on metrics like CLIP-score, PSNR, and DINO-features, which motivate us to use TMR-sim and FID metrics.** Despite these, we author team **quite respect your attitude on this** and recognize this is a limitation in the visual editing community.
>
> To make our results more convincing, we provided **almost all the experiments we could think of before the submission**. More importantly, we also package our method as **a web application** via gradio-app, whose video record is provided in the supplementary. All these efforts are to provide a holistic understanding of our method for readers.
>
> We always value the suggestions from reviewers significantly and are looking forward to further updates.
>
> Best,
>
> MotionCLR author(s) team

---

> ### Author Response · Authors · 2024-11-24
>
> Dear reviewer `b8bV`,
>
> Thanks for your prompt reply and engaging to improve the quality of this submission with us.
>
> ---
>
> We would like to answer your confusion on "sum-to-one" at first. This is because CLIP tokenizes a sentence into 77 tokens in all cases, including some placeholder tokens. In Fig.3-c, we only visualize the weight of "a", "person", and "jumps". As a result, the summation of three tokens is less than 1. If we visualize all 77 tokens, the figure will be not clear enough for presentation.
>
> We do understand that, as a reader, you may be confused that the manipulation will be on the attention map or similarity matrix. **We have thought about this before the submission.** Therefore, due to page limits, we provide `PyTorch`-like codes in Code-1 (Appendix F.1, page 35). Actually, our manipulation is on the attention map (after Softmax).
>
> *P.S.*: We guess that the reader will be curious about why we do not operate the similarity matrix directly. Answer: The value scales of the similarity matrix are without any guarantee (like "sum-to-one"), which is the motivation for using Softmax. Therefore, the adjusting weight in motion (de-)emphasizing is relatively hard to determine. With the sum-to-one property, it is much easier to determine the adjusting scale (like 0-1) for users. In this fashion, we can control the weight more precisely. You can refer to Fig.-14 (Appendix A.1, page 18). It is interesting that manipulation of the cross-attention maps will affect the number of "jump"s, additionally reflected by the self-attention maps in Fig.-14.
>
> ---
>
> We thank you again for joining such a detailed discussion. We are still open to answering your question before the deadline.
>
> Best,
>
> MotionCLR author(s) team

---

> ### Comment · Reviewer_b8bV · 2024-11-24
>
> The largest cross-attention weight for "jump" is approximately 0.05, as shown in Figure 3-(c). While this is about 5 times higher than the average weight (1/77 ≈ 0.013), this relatively small scale raises some concerns about the robustness of the word-frame correspondence. In this example, the combined attention weights for the three relevant tokens account for only 8% of the total attention at the peak frame, with the remaining 92% distributed across placeholder tokens. If the word-frame correspondence is indeed well-represented in the proposed architecture, should the attention weights for the sentence be larger?
>
> Let's suppose Figure 3-(c) represents a typical result of well-aligned word-frame correspondence. In contrast to the relatively small scale of the observed cross-attention weights in Figure 3, the magnitude of the weight adjustment in Table 2 (+0.30) appears disproportionately large. Adjusting the weight of a single token by +0.30 out of 77 candidates would allocate 40% of the total attention to that token, making it 23 times larger than the average attention weight. This significant discrepancy raises questions about whether such adjustments align with the model’s learned dynamics.
>
> Moreover, directly adjusting the attention weight risks violating the contraint of the sum-to-one. Especially, subtracting attention weights introduces a critical issue, as it can result in negative values. Considering that the attention map is a linear combination of token values constrained to a 77-dimensional simplex (where values range between 0 and 1 and the sum equals one), subtracting weights risks violating these constraints more severely than adding weights. Negative attention weights would push the model into an out-of-distribution state, as such values were never encountered during training, potentially leading to ill-posed outputs.
>
> Table 2 reports acceptable FID scores for weight adjustments in the range of -0.4 to +0.5. However, considering the scale of the cross attention map, I believe the anlaysis on weight adjustment between -0.3 ~ 0.3 is more important, but the table 2 lacks the results. Also, the evaluation is based on only 100 examples that were manually selected by the authors. This raises questions about the representativeness of the results. Given that the HumanML3D test dataset contains many single-verb examples, it would be helpful to clarify why the authors chose to manually construct a test set instead of conducting experiments on all single-verb examples in the dataset. This manual selection process reduces the generalizability of the experimental results.
>
> Additionally, although the authors mentioned that they considered the design choice for manipulating the similarity matrix, it would be beneficial if the rationale for this choice were more clearly explained in the paper or supplementary material, supported by quantitative evidence. Mentioning that this work represents early exploration in the text-to-motion domain does not fully address the lack of quantitative results. Providing a more thorough explanation and stronger supporting evidence would significantly strengthen the paper’s claims.
>
> In conclusion, I respectfully disagree with the authors' assertions, as the paper generally lacks sufficient quantitative results on its main contribution—word-frame correspondence—and the design choices for the subtasks. As reviewer cnao pointed out, the authors' assertion regarding word-frame correspondence appears to require further validation. I believe that word-frame correspondence should first be rigorously proven before being applied to various subtasks. However, the authors conducted experiments on subtasks and used the results as evidence for word-frame correspondence. I find this approach problematic, as all the experimental results are **indirect** evidence of word-frame correspondence and heavily rely on qualitative results. This critically weakens the reliability of the assertion and leads me to lean more toward clear rejection.
> (In the image domain, Liu et al.*[1] analyzed self-attention and cross-attention, using their quantitative analysis as evidence to support and justify their method.)
>
> Furthermore, the use of the term **explicit** modeling for the word-frame correspondence seems difficult to support, as no **explicit** guidance is provided at the word-frame level. For instance, in the image domain, Stable Diffusion employs cross-attention across modalities, but it would not typically be considered **explicit** modeling of region-word correspondence. To achieve fine-grained correspondence, models like GLIP have been proposed in the image domain. Similarly, I kindly suggest that the authors consider utilizing datasets with fine-grained labels, such as BABEL (although it is not a sentence-annotated dataset and requires preprocessing to generate sentence-level annotations), to establish truly **explicit**, fine-grained word-frame level modeling.

---

> > ### Comment · Reviewer_b8bV · 2024-11-24
> >
> > Reference
> >
> > [1] CVPR 2024, *Towards Understanding Cross and Self-Attention in Stable Diffusion for Text-Guided Image Editing*

---

> > > ### Author Response · Authors · 2024-11-26
> > > **Re: b8bV (1/2)**
> > >
> > > Dear reviewer `b8bV`,
> > >
> > > We acknowledge your response. We pay more efforts and add more experimental results to resolve your follow-up questions as follows.
> > >
> > > ---
> > >
> > > For the attention weights regarding "should the attention weights for the sentence be larger?", in the example of "a person jumps", the "jumps" is the key component to control the semantics in cross-attention. As a result, other words/tokens are relatively not as essential as this critical word "jumps". Besides, we think the $5\times$ weight over the average weights is relatively significant.
> > >
> > > We understand that editing the attention map directly will violate the sum-to-one property. As pointed by `hAdU`, we provide the suggested adjusting weight for users. In usage, the users can modify this weight interactively, which is shown in the interactive demo video. Considering the question about the reweighting weights between -0.3 and 0.3, we provide additional results here to include them as follows.
> > >
> > > |  | -0.3 | -0.2 | -0.1 | 0 | +0.1 | +0.2 | +0.3 |
> > > | --- | --- | --- | --- | --- | --- | --- | --- |
> > > | TMR-sim | 53.364 | 53.479 | 53.755 | 53.956 | 54.015 | 54.237 | 54.311 |
> > > | FID | 0.225 | 0.220 | 0.221 | 0.217 | 0.210 | 0.211 | 0.210 |
> > >
> > > These results are aligned with the provided results in the paper. The reason for not showing these results mainly due to the page limit. For the reason why we select the the test set manually, we treat the train and the test set of HumamML3D as the similar distribution. However, we would like to test our method in scenarios more aligned with the natural usage, which drives us to construct the test set manually.
> > >
> > > In our study before the submission, we found the manipulation directly on the similarity matrix is hard to control the manipulation scale precisely, which was stated in our previous reply. To validate our design choice of manipulating on the attention-map, rather than the similarity matrix, we add a experiment as follows.
> > >
> > > | manipulation | Metrics | -0.3 | -0.2 | -0.1 | 0 | +0.1 | +0.2 | +0.3 |
> > > | --- | --- | --- | --- | --- | --- | --- | --- | --- |
> > > | matrix | TMR-sim | 53.888 | 53.915 | 53.902 | 53.956 | 53.988 | 54.001 | 54.010 |
> > > | matrix | FID | 0.386 | 0.340 | 0.300 | 0.217 | 0.315 | 0.365 | 0.399 |
> > > | attention map | TMR-sim | **53.364** | **53.479** | **53.755** | **53.956** | **54.015** | **54.237** | **54.311** |
> > > | attention map | FID | **0.225** | **0.220** | **0.221** | **0.217** | **0.210** | **0.211** | **0.210** |
> > >
> > >
> > > As can be seen in the table, the manipulation of the similarity matrix cannot be controlled as well as on the attention map. Additionally, the manipulation will also affect the motion quality a bit, because the weights of all tokens will be manipulated jointly when manipulating. We do hope this qualitative comparison will strengthen the motivation for this design choice. We acknowledge your insightful suggestion on this to help us clarify the design choice.
> > >
> > > ---
> > >
> > > (next part as follows)

---

> > > > ### Author Response · Authors · 2024-11-26
> > > > **Re: b8bV (2/2)**
> > > >
> > > > We thank the in-depth discussion on metrics. We do understand your point on the qualitative results of the “word-motion correspondence”. To address your concerns, we conducted the following experiments (more detailed than previous response) to provide direct evidence.
> > > >
> > > > ---
> > > >
> > > > *Question*: **Is the cross-attention activation of the verb aligned with the action execution in the motion?**
> > > >
> > > > *Answer:*
> > > >
> > > > We treat the following result as the direct verification of the “word-motion correspondence”.
> > > >
> > > > We introduce the moment retrieval function in TMR, which can identify the action execution area in the motion. For example, for a motion generated by “a man walks forward”, we can calculate the similarity between the word “walks” and all 20-frame subintervals along the sequence. As a result, we can obtain the word-action similarity along the sequence. If the similarity value of the action is larger than 65% of the maximum attention values, we treat the action of the word is executed. Besides, if the attention value of the attention map is larger than 65% of the maximum attention value, we treat the word of action as activated in the attention map. Similarly, we also applied this activation assessment on the velocity and found the execution area of actions.
> > > >
> > > > Note that the velocity-based and moment-retrieval-based approaches introduced here are for a "double-check", both of which are approaches to determining the action execution in the motion domain.
> > > >
> > > > To verify whether the activation of word in attention is aligned with action execution, we calculate the IoU between (i) attention-activation and root-velocity-execution (*E1*), (ii) attention-activation and moment-retrieval-execution (*E2*).
> > > >
> > > > |  | adjusting weight | -0.1 | 0 | +0.1 |
> > > > | --- | --- | --- | --- | --- |
> > > > | *E1* | cross-attention & root velocity | 74.3% | 75.5% | 76.2% |
> > > > | *E2* | cross-attention & moment-retrieval-value | 72.5% | 73.2% | 73.1% |
> > > > | *E3* | root velocity & moment-retrieval-value | 76.5% | 77.2% | 77.4% |
> > > > | *E4* | neg. cross-attention & root velocity | 15.5% | 16.4% | 16.7% |
> > > > | *E5* | neg. cross-attention & moment-retrieval-value | 14.2% | 15.3% | 15.2% |
> > > >
> > > > As can be seen, compared with *E3* (both explicit action indicators: speed, moment retrieval of TMR),  the values in *E1*/*E2* are similar to those in *E3*. Therefore, the cross-attention activation is well aligned with motion execution.
> > > >
> > > > Moreover, we additionally provide two comparison groups (*E4*, *E5*). The setting is that, when calculating IoU, the cross-attention map for each example is replaced by an attention map of a negative example from the test set. **As can be seen (*E1* vs *E4*/*E5*), the cross attention is aligned with the action execution in the motion in the generation process of each motion.** We treat results can prove the word-motion correspondence in the method.
> > > >
> > > > For the claim of "explicit word-motion correspondence", we will revise this as the claim of "ine-grained word-motion correspondence". This is because we do not provide explicit supervision on this actually. Is the revised statement better than before? We are happy to learn your advice.
> > > >
> > > > If you have any suggestions to enhance the paper quality, we are always happy to follow.
> > > >
> > > > Thanks.
> > > >
> > > > MotionCLR author(s) team

---

> ### Comment · Reviewer_b8bV · 2024-11-27
> **Final recommendation**
>
> Thank you for providing additional experimental results. While I appreciate the authors’ efforts to address concerns, the justification for the manual construction of the test set is not particularly persuasive, and I do not believe that experimental results based on such a small, manually constructed test set are valid. The experiments in the reply are still based on a test set of only 100 samples, raising concerns about potential bias, as these samples may have been specifically constructed to support the authors' hypothesis. Additionally, the small size of the test set is insufficient to convincingly demonstrate the generalizability of their approach.
>
> It would be helpful to know if the test set includes long text descriptions and how it performs in such cases. What are the effects of longer texts? How many different verbs are present in the test set? Conducting experiments on the HumanML3D test dataset could provide a more robust and straightforward resolution to these concerns.
>
> Moreover, the design for adjusting the attention map lacks elaboration and does not account for the simplex property, further casting doubt on the validity of the attention map manipulation. Specifically:
>
> - The authors' explanation in L363 states, "Therefore, the suggested value of the weights ranges from −0.5 to +0.5."
> - There is an absence of experimental results in the range of -0.3 to 0.3 in the main paper.
> - There is a lack of deeper analysis on cross-attention mechanisms (e.g., what happens when attention weights have negative values? How is the violation of the simplex property handled?).
>
> These points suggest that the authors may not have fully accounted for the scale and behavior of the cross-attention map, as well as the implications of violating the simplex property. This raises concerns about whether the architectural design and the attention weight manipulation process were rigorously developed. Allowing users to adjust the weights does not fully address these issues.
>
> The table provided in the reply shows that manipulating attention scores performs better than manipulating the similarity matrix, but this conclusion is drawn solely within the context of the small test set. As mentioned in my previous comment and supported with a reference, the authors should evaluate the effects of manipulation not only on the targeted words but also on other words. A clear comparison demonstrating the effectiveness of word-frame correspondence is necessary to substantiate their claims.
>
> Furthermore, I find it difficult to agree with the architectural design being described as enabling "fine-grained and explicit word-frame correspondence". The authors neither utilize fine-grained annotations, such as those in the BABEL dataset, nor explicitly guide the training process to ensure fine-grained word-frame correspondence. Simply incorporating cross-attention does not guarantee fine-grained or explicit word-frame correspondence, and this does not qualify as a novel contribution of the paper.
>
> ## Conclusion
>
> While the authors have made efforts to address some of the concerns and I appreciate the messages the authors want to deliever to the community, the issues I raised remain significant. Addressing these concerns would require major revisions to the manuscript, including a re-evaluation of the paper's main contributions and the inclusion of more robust quantitative experiments to substantiate their claims. These changes are essential to strengthening the paper’s core arguments and logical structure. Given that ICLR is a top-tier conference on representation learning, the evidence provided does not yet meet the level of rigor required for acceptance. I believe the paper has strong messages if it's validated properly, so even though I can't agree with how to validate the assertion on this paper, I believe the paper will take important role once it clearly proves the correspondence.
>
> **Final Evaluation:**
> I would like to lower my evaluation of this paper from **5: marginally below the acceptance threshold** to **3: reject, not good enough**

---

### Official Review · Reviewer_DjgQ · 2024-11-03

**Soundness:** 2
**Presentation:** 2
**Contribution:** 3
**Rating:** 6
**Confidence:** 3

**Summary:**

The research presents MotionCLR, an attention-based motion diffusion model designed to enhance interactive editing of human motion generation. It addresses the shortcomings of previous models that struggle with word-level text-motion correspondence and lack explainability, limiting their fine-grained editing capabilities. MotionCLR utilizes self-attention to evaluate sequential similarities among motion frames and cross-attention to identify specific correspondences between word sequences and motion time steps. The authors present three main applications, each of which includes several specific variations that can be considered separate applications, as follows:
1. Motion emphasizing and de-emphasizing.
    - Motion erasing.
    - In-place motion replacement.
2. Motion sequence shifting.
    - Example-based motion generation.
3. Motion style transfer.

**Strengths:**

- The authors present many creative applications that utilize simple techniques, open up new applications for text-to-motion.
- MotionCLR effectively balances control and quality; instead of sacrificing one for the other, the model demonstrates improvements in both aspects.
- The editing can be performed during inference without the need for specific training for each task.
- The authors leverage existing datasets, allowing for more fine-grained control over motion generation.
- The paper is well-structured and easy to follow.

**Weaknesses:**

- This editing is out-of-distribution problem which may limit its performance in real-world applications. More qualitative and quantitative experiments would strengthen the claim of the model’s handling of out-of-distribution motions, providing clearer insights into its capabilities and limitations. (see next point)
- Lack of a comprehensive experiment discussion
  - It remains unclear how effectively the model can decouple complex actions, as text and motion data tend to be tightly coupled when there are limited samples available.
  - For example, if the dataset contains only one text-motion pair for "turns around and does a cartwheel", can MotionCLR generate a "cartwheel" without performing the prior action in the prompt, i.e. "turns around," using motion erasing (de-emphasizing)?
  - Additionally, let’s say the training set always shows "a man crouches forward" by starting with "standing up" before moving into the crouching position. If we apply "Motion sequence shifting" to place the "crouching" action at the beginning, can the model generate this motion without seeing this during training?

**Questions:**

In Table 1, is the generation quality is improving from other diffusion models? ("ReMoDiffuse" is retrieval method and MoMask is using masked modeling). If yes, I think this advantage can be emphasis too.

---

> ### Author Response · Authors · 2024-11-17
>
> Reviewer `DjgQ` provides us with constructive suggestions to discuss the limitations and the emphasis on improvement. We authors appreciate this sincerely and will add `DjgQ` to our acknowledgment if accepted.
>
> ---
>
> **Q1**: About OOD cases.
>
> **A1**: We acknowledge our work will related to some OOD motions. Therefore, we author team is now collecting the motion data to enlarge our database and **train our V2 model** on the **larger dataset**. We hope our efforts will resolve this issue.
>
> **Q2.1**: More complex motions.
>
> **A2.1**: As shown in Fig. 7, our method can accurately handle a motion sequence with actions more than one. We also provide the attention map visualization of a complex motion in Appendix D (Fig. 26). The results show the good localization ability of attention map in a complex motion.
>
> **Q2.2**: Test case issues and limitation discussion.
>
> **A2.2**: Thanks for helping us enhance the exploration of the extreme test of our method. We test these cases of limited training samples. We trained the model as you suggested and found the generated results almost similar to the original unedited motion. The reason mainly comes from the limited cases the model has seen during training. **As a result, the model cannot distinguish “cartwheel” and “turns”, because the model learns to memorize limited examples in training.** That is to say, the model does not know what motion is not a “cartwheel”, thereby restricting the controllability of attention conditions. **We included this in our limitation part of PDF.**
>
> **Q3**: Improved baseline comparison.
>
> **A3**: Your suggestion is quite insightful! This work is based on MDM and MotionDiffuse. The direct comparison is as follows.
>
> | method | Top 1 | Top 2 | Top 3 | FID | MM-Dist | Multi-Modality |
> | --- | --- | --- | --- | --- | --- | --- |
> | MDM | - | - | 0.611 | 0.544 | 5.566 | **2.799** |
> | MotionDiffuse | 0.491 | 0.681 | 0.782 | 0.630 | 3.113 | 1.553 |
> | MotionCLR | **0.542** | **0.733** | **0.827** | **0.099** | **2.981** | 2.145 |
>
> As our work introduces explicit modeling on word-level correspondence, it significantly improves the generation quality than baselines. **We thank you for this constructive question, and will include you in our acknowledgement if accepted.**
>
> ---
>
> If you have no further concerns, could you please consider **reevaluating this work and improving your rating**? If not or if you have further guidance, we are happy to resolve them.

---

> > ### Author Response · Authors · 2024-11-20
> >
> > Dear Reviewer `DjgQ`:
> >
> > Thanks again for your efforts in reviewing. We are looking forward to your reply. Would you mind checking our response and confirming if there are unclear explanations?
> >
> > We are highly encouraged if your concerns have been addressed. If your concerns are resolved, could you please consider reevaluating this work and improving your rating? If you need any more clarification, we can provide it as soon as possible before the discussion deadline.
> >
> > Best,
> >
> > MotionCLR author(s) team

---

> > > ### Author Response · Authors · 2024-11-23
> > >
> > > Dear Reviewer `DjgQ`:
> > >
> > > Thanks again for your efforts in reviewing. We are looking forward to your reply. Would you mind checking our response and confirming if there are unclear explanations?
> > >
> > > We are highly encouraged if your concerns have been addressed. **If your concerns are resolved, could you please consider reevaluating this work and improving your rating?** If you need any more clarification, we can provide it as soon as possible before the discussion deadline.
> > >
> > > Best,
> > >
> > > MotionCLR author(s) team

---

> > > > ### Comment · Reviewer_cnao · 2024-11-23
> > > >
> > > > Thank you for the author’s response and the additional experimental results. They address some of my concerns, such as the generalizability of the proposed work on human motion generation from text. However, my concern regarding the temporal correspondence between words and motions still remains. While absolute positional encoding provides positional information, it only offers ‘unique’ positional markers, unlike relative positional encoding, which captures sequential order information. Without this sequential information or additional techniques (e.g., CTC, masking) to guide the alignment between sequences (words and motions), the model struggles to learn the correspondence between the two. This is particularly concerning given that HumanML3D is not large enough to enable the network to learn such correspondence solely from the provided sequences, without explicit alignment information (e.g. per frame action label). For instance, if the order of action words changes, the network cannot infer these changes. Consequently, editing specific words in the sequence might alter the entire motion rather than just the corresponding parts, as the network lacks a mechanism to control this correspondence.
> > > >
> > > > Moreover, absolute positional encoding is vulnerable to sequence length, especially in long-term generation, which limits its applicability. I also agree with reviewer `b8bV` that the paper lacks proper quantitative evaluation to demonstrate temporal correspondence between action words and the generated motion. Without a thorough evaluation in this context, it is difficult to trust that the network performs well for fine-grained motion editing. Therefore, I am maintaining my original rating.

---

> > > > > ### Author Response · Authors · 2024-11-24
> > > > >
> > > > > Dear reviewer `cnao`,
> > > > >
> > > > > Thanks for your reply. We are happy that some of your concerns are resolved.
> > > > >
> > > > > We know that one of your concerns is about our design choice of absolute positional encoding (APE) *v.s.* relative positional encoding (RPE). However, APE is still capable of modeling the order of sequence. To our understanding, the CTC and masking mechanisms are approaches to enhance the sequential modeling, which cannot indicate APE is not effective. Our visualization of both cross-attention and self-attention in Fig. 3/26/27 also verifies our claim. We are not clear why APE cannot model the sequential order information. Could you please provide more information about this?
> > > > >
> > > > > Considering the question of positional encoding of text, we follow the common setting of CLIP, which is also used by most motion generation methods like MDM, MotionDiffuse, T2M-GPT. The CLIP has a maximum number of tokens as 77. Therefore, it will never suffer the scenario of variable sequence length.
> > > > >
> > > > > Our key insight exactly lies in how to model the word-motion correspondence without external supervision, but just with the cross-attention modeling. **We also notice that the CLIP-based image generation also shows the word-level correspondence between each word and the motion, such as [MasaCtrl](https://arxiv.org/pdf/2304.08465) (ICCV-23), [Cross-Image Attention](https://arxiv.org/pdf/2311.03335) (SIGGRAPH-24), [Prompt-to-Promt](https://openreview.net/forum?id=_CDixzkzeyb) (ICLR-23, mentioned by `hAdU`). Although all these methods use the APE without external labels, the fine-grained correspondence has been widely verified by attention-based image editing methods. We are still not clear why APE is not effective to model such correspondence.** If you can provide more information, we are happy to learn from that.
> > > > >
> > > > > We kindly respect your opinions and welcome for your reply!
> > > > >
> > > > > Best,
> > > > >
> > > > > MotionCLR author(s) team

---

> > > > > > ### Comment · Reviewer_cnao · 2024-11-25
> > > > > >
> > > > > > Thank you for your response. I want to clarify that RPE, CTC, and masking are examples that can strengthen the sequential correspondence. I am not suggesting that these modules are strictly necessary, but without a design specifically aimed at reinforcing word-motion correspondence, simply relying on cross-attention does not guarantee such correspondence, as the authors imply. Furthermore, the authors provide examples of motion editing, such as in Figures 3, 26, and 27, but these are not quantified over the test set. Without quantitative results or architectural designed to enhance correspondence, it is difficult to accept that cross-attention alone improves correspondence. Additionally, the papers referenced pertain to the image-language domain, which typically benefits from large corpora of data. As mentioned in my previous comments, I do not believe that HumanML3D provides a sufficiently large dataset for the network to learn the correspondence between motion and words without explicit guidance.

---

> > > > > > > ### Author Response · Authors · 2024-11-26
> > > > > > > **To DjgQ: Looking forward to your reply**
> > > > > > >
> > > > > > > Dear Reviewer `DjgQ`:
> > > > > > >
> > > > > > > Thanks again for your efforts in reviewing. We are looking forward to your reply. Would you mind checking our response and confirming if there are unclear explanations?
> > > > > > >
> > > > > > > We are highly encouraged if your concerns have been addressed. If your concerns are resolved, could you please consider reevaluating this work and improving your rating? If you need any more clarification, we can provide it as soon as possible before the discussion deadline.
> > > > > > >
> > > > > > > Best,
> > > > > > >
> > > > > > > MotionCLR author(s) team

---

> > > > > > > > ### Comment · Reviewer_DjgQ · 2024-11-27
> > > > > > > > **Official Comment by Reviewer DjgQ**
> > > > > > > >
> > > > > > > > Thank you for clarifying my concern.
> > > > > > > > Given that HumanML3D is a relatively small dataset for the model to learn fine-grained relationships between descriptions and motions, the proposed method faces challenges with OOD, which might limit its applicability in real-world applications. Additionally, I agree with the concern raised by reviewer ```b8b``` that the evaluation may not be sufficient to comprehensively measure motion editing. **However, I still find the proposed method novel and creative. I lean toward accepting, so I keep my borderline accept rating.**
> > > > > > > >
> > > > > > > >
> > > > > > > > **Questions**
> > > > > > > >
> > > > > > > > 1. Why not use "DDIM" as a baseline? As shown in Table 1, the FID score seems significantly better than the DPM-solver version.
> > > > > > > > 2. Is the baseline from Table 2. the same as "MotionCLR*" from Table 1. If so why FID score are different?
> > > > > > > >
> > > > > > > > Typo: Border impact statement -> Broader

---

> ### Author Response · Authors · 2024-11-24
> **To DjgQ: Looking forward to your reply**
>
> Dear Reviewer `DjgQ`:
>
> Thanks again for your efforts in reviewing. We are looking forward to your reply. Would you mind checking our response and confirming if there are unclear explanations?
>
> We are highly encouraged if your concerns have been addressed. **If your concerns are resolved, could you please consider reevaluating this work and improving your rating?** If you need any more clarification, we can provide it as soon as possible before the discussion deadline.
>
> Best,
>
> MotionCLR author(s) team

---

> ### Author Response · Authors · 2024-11-27
>
> Dear reviewer `DjgQ`,
>
> Thanks for your response and **emphasizing our method is novel and creative**! **We quite acknowledge your effort to discuss with us and so detailed check.**
>
> 1. Thanks for such detailed checking. For the DDIM/DPM-solver design choice, we treat both choices are acceptable/comparable, and both choices can be used in the inference stage. Besides, the R-Precision metric between DDIM and DOM-solver is comparable. As a result, we do not specify the default choice deliberately.
> 2. Thanks for this discussion. Because we need to annotate the words that should be edited in the sentence, we *manually* construct the evaluation set for this, following PnP [1] in the image editing domain. Because of the difference in evaluation sets, the results in Table 1 and 2 are not the same.
> 3. For the typo, we will revise it accordingly in the manuscript.
>
> Your insightful comments help us to polish the paper a lot. For the fine-grained correspondence between cross-attention and the the final motion, we provide direct evidence to respond `b8bV` via the IoU metrics between cross-attention and the the final motion execution. This shows the attention activation is well aligned with the final motion.
>
> [1]: Tumanyan, N., Geyer, M., Bagon, S., & Dekel, T. (2023). Plug-and-play diffusion features for text-driven image-to-image translation. In *Proceedings of the IEEE/CVF Conference on Computer Vision and Pattern Recognition* (pp. 1921-1930). **Citation:540**
>
> ---
>
> We paste the evidence to respond to `b8bV` via the IoU metrics as follows. If you have any suggestion to help us improve the evaluation, we are very very grateful.
>
> *Question*: **Is the cross-attention activation of the verb aligned with the action execution in the motion?**
>
> *Answer:*
>
> We treat the following result as the direct verification of the “word-motion correspondence”.
>
> We introduce the moment retrieval function in TMR, which can identify the action execution area in the motion. For example, for a motion generated by “a man walks forward”, we can calculate the similarity between the word “walks” and all 20-frame subintervals along the sequence. As a result, we can obtain the word-action similarity along the sequence. If the similarity value of the action is larger than 65% of the maximum attention values, we treat the action of the word is executed. Besides, if the attention value of the attention map is larger than 65% of the maximum attention value, we treat the word of action as activated in the attention map. Similarly, we also applied this activation assessment on the velocity and found the execution area of actions.
>
> Note that the velocity-based and moment-retrieval-based approaches introduced here are for a "double-check", both of which are approaches to determining the action execution in the motion domain.
>
> To verify whether the activation of word in attention is aligned with action execution, we calculate the IoU between (i) attention-activation and root-velocity-execution (*E1*), (ii) attention-activation and moment-retrieval-execution (*E2*).
>
> |  | adjusting weight | -0.1 | 0 | +0.1 |
> | --- | --- | --- | --- | --- |
> | *E1* | cross-attention & root velocity | 74.3% | 75.5% | 76.2% |
> | *E2* | cross-attention & moment-retrieval-value | 72.5% | 73.2% | 73.1% |
> | *E3* | root velocity & moment-retrieval-value | 76.5% | 77.2% | 77.4% |
> | *E4* | neg. cross-attention & root velocity | 15.5% | 16.4% | 16.7% |
> | *E5* | neg. cross-attention & moment-retrieval-value | 14.2% | 15.3% | 15.2% |
>
> As can be seen, compared with *E3* (both explicit action indicators: speed, moment retrieval of TMR),  the values in *E1*/*E2* are similar to those in *E3*. Therefore, the cross-attention activation is well aligned with motion execution.
>
> Moreover, we additionally provide two comparison groups (*E4*, *E5*). The setting is that, when calculating IoU, the cross-attention map for each example is replaced by an attention map of a negative example from the test set. **As can be seen (*E1* vs *E4*/*E5*), the cross attention is aligned with the action execution in the motion in the generation process of each motion.** We treat results can prove the word-motion correspondence in the method.
>
>
> Thanks.
>
> MotionCLR author(s) team

---

### Official Review · Reviewer_cnao · 2024-11-03

**Soundness:** 3
**Presentation:** 4
**Contribution:** 2
**Rating:** 3
**Confidence:** 5

**Summary:**

This work introduces MotionCLR, a model designed for versatile human motion generation tasks based on textual input. MotionCLR leverages an attention mechanism to align text with motion, allowing it to handle tasks like in-place motion replacement and motion generation from text without retraining. A key focus is on analyzing how the attention mechanism facilitates alignment between text and motion, enabling effective generation for various tasks. Experimental results demonstrate the method’s effectiveness across a range of motion generation scenarios.

**Strengths:**

- The paper clearly presents the methodology and contributions, with a well-organized structure that makes it easy to follow.
- Figures are well-designed and effectively illustrate the approach and context.
- Detailed appendices provide additional information that supports readers’ understanding of the work.
- The authors conduct a wide variety of experiments, showcasing the advantages of MotionCLR across different tasks.

**Weaknesses:**

1. CLR Module Design: The design of the CLR module lacks detailed explanation. Specifically, the paper doesn’t clarify how MotionCLR differs from existing approaches or justify the use of U-Net-style transformer stacks for downsampling and upsampling time dimensions. Additional experiments or analyses are necessary to justify this architectural choice over replacing input condition style to cross attention-style from existing transformer-diffusion models.

2. Dataset Generalizability: Experiment in Sec 5.1 is conducted solely on the HumanML3D dataset for text-based motion generation. To validate generalizability across datasets, additional experiments on the KIT dataset are recommended.

3. Section 3.3 Relevance: Section 3.3 mainly covers widely known concepts without introducing new insights. Condensing it or merging it with Section 3.2 could improve clarity and conciseness.

4. Cross-Attention Explanation: More detailed explanations or illustrations are needed on how cross-attention aligns text with motion. For example:
- How is positional information of the text inserted? Is a positional embedding used?
- How does the network align text with motion tokens, given that text tokens doesn’t inherently contain sequential order information? I cannot find any details for alignment since this is critical when aligning two different sequences. Relative positional embedding or CTC like loss may be required to sequentially align motion and text tokens. Please clarify this point.

5. Longer Text Testing: Testing MotionCLR on longer text inputs with multiple action verbs (e.g., >5 action words) would offer a more practical assessment of its performance in more practical scenarios.

**Questions:**

Please refer to weaknesses.

---

> ### Author Response · Authors · 2024-11-17
>
> We sincerely acknowledge your contribution to handling this submission and provide constructive suggestions for this work. We will add you to our acknowledgment if accepted.
>
> ---
>
> **Q1**: About CLR module design.
>
> **A1**: Thanks for helping us enhance the clarity of this paper! We would like to answer this question from two aspects.
>
> - **CLR module design.** The main difference is that the CLR module models the correlation between the motion and each word and separates the timestep injection with the text condition. There is no previous work models text-motion so fine-grained and observe the explainability between each word and the motion sequence. We additionally provide an ablation on w/o down/up-sampling below. The results are similar to that of HumanML3D. The relatively smaller computation in down/up-sampling encourages us to set up this design choice.
>
>
>     | method | Top 1 | Top 2 | Top 3 | FID | MM-Dist | Multi-Modality |
>     | --- | --- | --- | --- | --- | --- | --- |
>     | w/o down/up-sampling | 0.540 | 0.731 | 0.818 | 0.101 | 2.984 | **2.152** |
>     | MotionCLR | **0.542** | **0.733** | **0.827** | **0.099** | **2.981** | 2.145 |
> - **About replacing input condition style to cross attention style.** The key insight of introducing cross-attention controlling is modeling word-level correspondence with motion, which supports our motion editing applications. The comparison with previous fashions is detailed in Appendix C.3 of the original submission. Besides, the ablation is also provided in Sec. 5.3.
>
> **Q2**: Results on KIT dataset.
>
> **A2**: Thanks for discussing the dataset generalizability! We provide our experiments on the largest text2motion dataset HumanML3D with 44970 texts. The KIT-ML is much smaller than HumanML3D, with 6278 texts. **We think the results on a larger database are more convincing. Despite these, we also provide additional experiments on the KIT dataset as follows.** The conclusion is similar to the result of HumanML3D.
>
> | method | Top 1 | Top 2 | Top 3 | FID | MM-Dist | Multi-Modality |
> | --- | --- | --- | --- | --- | --- | --- |
> | MoMask (SOTA) | 0.433 | 0.656 | 0.781 | **0.204** | 2.779 | 1.131 |
> | MotionCLR | **0.438** | **0.658** | **0.783** | 0.275 | **2.773** | **1.213** |
>
> **Q3**: About Section 3.3 relevance.
>
> **A3**: We acknowledge you for helping enhance the readability of the paper! Sec. 3.3 is critical to our understanding of cross-/self-attention in motion generation, which supports our versatile downstream tasks, especially Remark 1 and Remark 2. Considering your concern, we agree to merge them with Sec. 3.2 to enhance the readability.
>
> **Q4**: Positional embedding in cross-attention.
>
> **A4**: Thanks for discussing this! We treat the positional embedding of CLIP embeddings as a common setting. Therefore, we did not specify this in the original text. The positional embeddings are added to the text directly as most methods do. Thanks for pointing it out. Here is the PyTorch codes.
>
> ```python
> def encode_text(self, raw_text, device):
>     texts = clip.tokenize(raw_text, truncate=True) # [bs, context_length]
>     x = self.clip_model.token_embedding(texts).type(self.clip_model.dtype)  # [bs, n_ctx, d_model]
>     x = x + self.clip_model.positional_embedding.type(self.clip_model.dtype)# add position emb.
> ```
>
> **Q5**: Long text test.
>
> **A5**: Good question! We do not find the text with more than 5 words in the HumanML3D dataset, due to the limited number of such samples. Besides, the limited duration of less than 10 seconds in the dataset cannot contain such complex motions. We provide several visualizations of longer text inputs with multiple action verbs in Appendix A.5 of the original submission. Besides, results in Appendix D also verify the explainability of the case with multiple actions.
>
> ---
>
> Dear reviewer `cnao`, if your concerns are resolved, could you please consider reevaluating this work and improving your rating? If not or if you have further guidance, we are happy to resolve them.

---

> > ### Author Response · Authors · 2024-11-20
> >
> > Dear Reviewer `cnao`:
> >
> > Thanks again for your efforts in reviewing. We are looking forward to your reply. Would you mind checking our response and confirming if there are unclear explanations?
> >
> > We are highly encouraged if your concerns have been addressed. If your concerns are resolved, could you please consider reevaluating this work and improving your rating? If you need any more clarification, we can provide it as soon as possible before the discussion deadline.
> >
> > Best,
> >
> > MotionCLR author(s) team

---

> > > ### Author Response · Authors · 2024-11-23
> > >
> > > Dear Reviewer `cnao`:
> > >
> > > Thanks again for your efforts in reviewing. We are looking forward to your reply. Would you mind checking our response and confirming if there are unclear explanations?
> > >
> > > We are highly encouraged if your concerns have been addressed. **If your concerns are resolved, could you please consider reevaluating this work and improving your rating?** If you need any more clarification, we can provide it as soon as possible before the discussion deadline.
> > >
> > > Best,
> > >
> > > MotionCLR author(s) team

---

> ### Author Response · Authors · 2024-11-26
>
> Dear reviewer `cnao`,
>
> Thanks for your reply. We do understand your point that the mentioned RPE/CTC methods may enhance the result. We also agree the relative task will benefit from a larger-scale of motion database. However, the HumanML3D is the largest text2motion dataset that can accessed. We do believe our method will be more scalable when obtaining a large database, which is also our future work. Besides, if you have more replies, could they be posted under your review? We are worried that the current replies will be confused with `DjgQ`'s replies. Thanks a lot!
>
> Best,
>
> MotionCLR author(s) team

---

> ### Comment · Reviewer_cnao · 2024-11-27
>
> Thank you for moving our discussion to the correct location. I mistakenly added my comment in the wrong place earlier. After interacting with the author and considering the feedback from reviewer `b8bV`, I also lean towards recommending rejection of this work.
>
> While I acknowledge that this paper presents an earlier attempt at motion editing focusing on cross-attention between motion and words, and the overall quality is commendable with diverse experiments and analyses, it falls short in a critical aspect: the lack of quantitative evaluation for the proposed method. This omission makes it difficult to assess the true impact of the work. As the author mentioned, this work might perform well for motion editing without a specific design for motion-word correspondence. However, it is difficult to substantiate the author’s claim based solely on qualitative results. Additionally, the absence of such evaluations leaves the claims of the paper unsupported and raises concerns about reproducibility and practical relevance. I am confident that with proper quantitative evaluations, this paper would likely reach the level needed for acceptance. However, in its current form, I cannot consider it acceptable. I recommend that the authors incorporate quantitative evaluations and resubmit the work. Accordingly, I am changing my rating from borderline reject to reject.

---

> > ### Author Response · Authors · 2024-11-27
> >
> > Dear reviewer `cano`,
> >
> > Thanks for your reply and the acknowledgment of the overall quality. To resolve the concerns regarding the evaluation, we make much effort to resolve them.
> >
> > We paste the evidence to respond to `b8bV` via the IoU metrics as follows, which we think is enough to support the correspondence. If you have any suggestions to help us improve the evaluation or specify the quantitative metric, we are very very grateful.
> >
> > ---
> >
> > *Question*: **Is the cross-attention activation of the verb aligned with the action execution in the motion?**
> >
> > *Answer:*
> >
> > We treat the following result as the direct verification of the “word-motion correspondence”.
> >
> > We introduce the moment retrieval function in TMR, which can identify the action execution area in the motion. For example, for a motion generated by “a man walks forward”, we can calculate the similarity between the word “walks” and all 20-frame subintervals along the sequence. As a result, we can obtain the word-action similarity along the sequence. If the similarity value of the action is larger than 65% of the maximum attention values, we treat the action of the word is executed. Besides, if the attention value of the attention map is larger than 65% of the maximum attention value, we treat the word of action as activated in the attention map. Similarly, we also applied this activation assessment on the velocity and found the execution area of actions.
> >
> > Note that the velocity-based and moment-retrieval-based approaches introduced here are for a "double-check", both of which are approaches to determining the action execution in the motion domain.
> >
> > To verify whether the activation of word in attention is aligned with action execution, we calculate the IoU between (i) attention-activation and root-velocity-execution (*E1*), (ii) attention-activation and moment-retrieval-execution (*E2*).
> >
> > |  | adjusting weight | -0.1 | 0 | +0.1 |
> > | --- | --- | --- | --- | --- |
> > | *E1* | cross-attention & root velocity | 74.3% | 75.5% | 76.2% |
> > | *E2* | cross-attention & moment-retrieval-value | 72.5% | 73.2% | 73.1% |
> > | *E3* | root velocity & moment-retrieval-value | 76.5% | 77.2% | 77.4% |
> > | *E4* | neg. cross-attention & root velocity | 15.5% | 16.4% | 16.7% |
> > | *E5* | neg. cross-attention & moment-retrieval-value | 14.2% | 15.3% | 15.2% |
> >
> > As can be seen, compared with *E3* (both explicit action indicators: speed, moment retrieval of TMR),  the values in *E1*/*E2* are similar to those in *E3*. Therefore, the cross-attention activation is well aligned with motion execution.
> >
> > Moreover, we additionally provide two comparison groups (*E4*, *E5*). The setting is that, when calculating IoU, the cross-attention map for each example is replaced by an attention map of a negative example from the test set. **As can be seen (*E1* vs *E4*/*E5*), the cross attention is aligned with the action execution in the motion in the generation process of each motion.** We treat results can prove the word-motion correspondence in the method.
> >
> > Best,
> >
> > MotionCLR author(s) team

---

### Author Response · Authors · 2024-11-17
**Warm-up Discussion**

Dear AC, SAC, and all reviewers,

We author team of MotionCLR sincerely appreciate your contributions to handling this submission and helping us polish the quality of this work. We are delighted that reviewers generally acknowledge the following aspects of this work.

- **Open up new effective applications** and **playing an important role** for text-to-motion.
- **Clear** presentations and detailed information/experiments.

Besides, some reviewers also highlight other strengths:

- **Novel** contribution and new understanding. (**hAdU**, **b8bV**, **DjgQ**)
- **Impressive** editing results and **effective** generation performance. (**hAdU**, **DjgQ**, **cnao**)

As this is an **early exploration** of interactive motion editing and understanding attention in motion generation, we try to resolve the concerns from reviewers in corresponding parts and the revised parts have been highlighted as blue in PDF.

Best,

MotionCLR author(s) team

---

> ### Author Response · Authors · 2024-11-18
> **Follow-up discussion**
>
> Dear reviewers,
>
> Thanks again for your constructive suggestions! As the discussion period is quite tight, we would like to send you a kind reminder. We were wondering whether you had the chance to look at our response and whether there is anything else you would like us to clarify.
>
> We sincerely hope that our response regarding your concerns will be taken into consideration. **If you have no further concerns, could you please reevaluate this work and consider improving your rating?**  If not, please let us know and we remain open and would be more than glad to actively discuss them with you.
>
> Best,
>
> MotionCLR author(s) team

---

> > ### Author Response · Authors · 2024-11-22
> > **Follow-up discussion**
> >
> > Dear reviewers,
> >
> > Thanks again for your constructive suggestions! As the discussion deadline is approaching, we would like to send you a kind reminder. We were wondering whether you had the chance to look at our response and whether there is anything else you would like us to clarify.
> >
> > We sincerely hope that our response regarding your concerns will be taken into consideration. If you have no further concerns, could you please reevaluate this work and consider improving your rating? If not, please let us know and we remain open and would be more than glad to actively discuss them with you.
> >
> > Best,
> >
> > MotionCLR author(s) team

---

### Meta-Review · Area_Chair_q8vn · 2024-12-21

**Metareview:**

This paper proposed a motion diffiusion model to support interactive editing of human motion generation.  The key contributions center around attention mechanisms to support CLR-based learning.  Effectiveness is demonstrated on three applications: motion editing, sequence shifting and style transfer.

The reviewers recognize the interesting nature of the work presented, and the creative applications that can result.

The reviewers point to several weaknesses, including: clear presentation, weak motivation and link to design and most importantly, insufficient experimental verification.  The last point on experimental verification is especially problematic and led to extensive discussion between the reviewers after the author exchange period.

After closed discussions, this paper received three rejects and one positive review (hAdU).  The initially positive reviewers DjgQ and hAdU all downgraded their reviews.

After reading through the paper, reviews and author responses, the AC proposes to reject the paper.  While the idea and results are interesting, the paper is not yet ready for publication.  In particular, stronger experimental results on a larger test set and or better qualified benchmark is necessary to verify the effectiveness.  Additionally, the paper would benefit from revisions to improve the overall writing in terms of method and motivation.

**Additional Comments On Reviewer Discussion:**

DjgQ states in discussion that they agree with reviewers b8bV and cnao regarding the weak evaluation (only 100 samples with repeated text), which the authors themselves acknowledge as a limitation of their own work.

Note that the review from hAdU is downweighted, due to reviewer inexperience - they state this is their first time reviewing for ICLR and are unsure about the bar for ICLR.  They remain positive about the work, but state that a more appropriate score for the work sits between 6 and 8.

---

### Decision · Program_Chairs · 2025-01-22

Reject